# POSE PRIOR LEARNER: UNSUPERVISED CATEGORICAL PRIOR LEARNING FOR POSE ESTIMATION

**Ziyu Wang    Shuangpeng Han    Mengmi Zhang**
Deep NeuroCognition Lab, Nanyang Technological University, Singapore
Address correspondence to `mengmi.zhang@ntu.edu.sg`

## ABSTRACT

A prior represents a set of beliefs or assumptions about a system, aiding inference and decision-making. In this paper, we introduce the challenge of unsupervised categorical prior learning in pose estimation, where AI models learn a general pose prior for an object category from images in a self-supervised manner. Although priors are effective in estimating pose, acquiring them can be difficult. We propose a novel method, named Pose Prior Learner (PPL), to learn a general pose prior for any object category. PPL uses a hierarchical memory to store compositional parts of prototypical poses, from which we distill a general pose prior. This prior improves pose estimation accuracy through template transformation and image reconstruction. PPL learns meaningful pose priors without any additional human annotations or interventions, outperforming competitive baselines on both human and animal pose estimation datasets. Notably, our experimental results reveal the effectiveness of PPL using learned prototypical poses for pose estimation on occluded images. Through iterative inference, PPL leverages the pose prior to refine estimated poses, regressing them to any prototypical poses stored in memory. Our code, model, and data are publicly available at: link.

## 1 INTRODUCTION

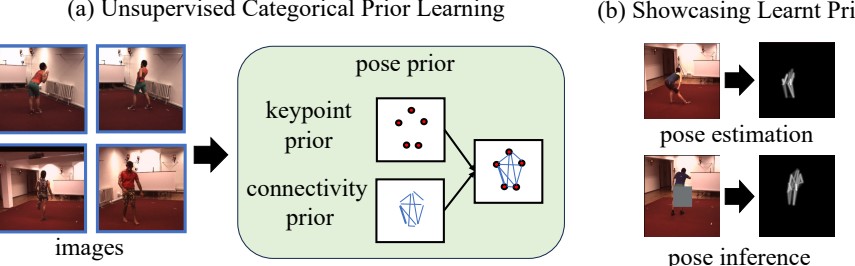

Figure 1: **Schematic illustration of the unsupervised categorical prior learning challenge.** On the left (a), given a series of input images (blue frames), the goal is to learn a pose prior (green rectangle) in a fully self-supervised manner. This pose prior consists of both keypoint and connectivity priors. To tackle this problem, we introduce the model, named as Pose Prior Learner (PPL). **Demonstration of the learned prior in pose estimation and inference under occlusion scenarios.** On the right (b), we showcase how the learned pose prior enhances performance in challenging tasks such as pose estimation and body pose inference under occlusion. Notably, although the PPL are trained only on full-body, non-occluded images, our proposed approach is still able to produce plausible pose predictions even when substantial occlusions are present.

Priors represent beliefs or assumptions about a system or the characteristics of a concept. They are widely used in statistical inference Lindley (1961), cognitive science Schad et al. (2021), and machine learning Diligenti et al. (2017); Gülçehre & Bengio (2016). This pre-existing knowledge is essential for guiding the inference process, enabling AI models to make robust predictions in uncertain or ambiguous situations Thiruvenkadam et al. (2008); Sung et al. (2015); Liang et al. (2024). The objective of our work is to enhance our understanding of priors in AI models and offer preliminary

answers to the three key intelligence questions: (1) How do we acquire priors in the first place? (2) Can we learn them from input data in a self-supervised manner? (3) Can we enhance the quality of the priors? To address these questions, we first formalize the challenge of unsupervised categorical prior learning, propose a computational model with hierarchical memory to learn such priors, and then use pose estimation as a testbed to evaluate the quality of the learned priors. See **Figure 1** for the schematic illustration of the challenge. Categorical pose estimation is a classical computer vision task that identifies the structure of objects belonging to the same category by detecting their keypoints. A pose prior summarizes the common characteristics shared by a variety of poses. It encapsulates the expectation of the keypoint configurations and the connectivity between keypoints.

In parallel to our challenge of unsupervised categorical prior learning from images for pose estimation, unsupervised pose estimation leverages the abundant, unannotated visual information available in large image datasets to extract pose information Hu & Ahuja (2021); Sommer et al. (2024b); Chen et al. (2019); He et al. (2022a); Schmidtke et al. (2021). The use of pose priors can provide valuable guidance in this process. We categorize the existing works in unsupervised pose estimation into two groups: those that incorporate hand-made priors and those that operate without any priors.

Recent approaches He et al. (2022a); Sun et al. (2022; 2023) attempt to predict keypoints from images, construct object structure representations using these keypoints, and learn effective structural information through image reconstruction. However, without pose priors, these methods can be disrupted by background information or may predict infeasible topological configurations of an object during occlusion. The risk of generating inaccurate keypoints stems from the absence of supplementary information that could help refine both keypoint localization and the connections between them.

The other group of methods Schmidtke et al. (2021); Yoo & Russakovsky (2023) utilize prior knowledge of a category's general pose to guide the pose estimation of individuals within that category. Conceptually, each category is expected to exhibit a generalized and distinctive pose prior that reflects characteristics such as shape, size, and structure. Individual poses should be seen as geometric transformations of this category-specific pose prior. As a result, employing a category-specific pose prior aids in guiding and regularizing the learning of poses. However, obtaining comprehensive general pose priors is highly challenging, as it requires extensive human annotations, particularly for novel categories. Moreover, human annotations may introduce implicit biases, hindering models from learning more meaningful priors.

Loosely inspired by how humans develop a general prior representation of an object category by observing individual object instances in images and subsequently using them to infer upcoming individual poses, we propose a new method called the Pose Prior Learner (PPL). PPL is designed to effectively learn a meaningful pose prior for a certain object category. It utilizes a hierarchical memory to store a finite set of prototypical poses and extract a general pose prior from them. Initially, both the hierarchical memory and the prior are randomly initialized but learnable parameters. During training, effective pose learning is supervised through image reconstruction. As training progresses, the hierarchical memory retains and aggregates multiple accurate prototypical poses, thereby contributing to a more precise pose prior and enhancing the model's ability to estimate poses. Similar to Wu et al. (2023b) and Li et al. (2024), PPL learns object poses via image-level reconstruction, but uniquely does so from unannotated images without any object masks. Moreover, although methods such as Sommer et al. (2024a) also tackle category-level pose estimation, they rely on dense point clouds for scene reconstruction. In contrast, PPL leverages sparse keypoints and their relations, yielding a lightweight, semantic, and invariant pose representation well-suited for high-level scene understanding.

While existing unsupervised pose estimation methods may implicitly encode structural priors, these priors are buried within the network parameters and remain opaque and non-interpretable. Such approaches learn pose priors only as latent model weights, without exposing their underlying structure. In contrast, our method explicitly extracts pose priors and represents them through symbolic keypoint and connectivity priors. This explicit representation is deterministic, structured, and interpretable, allowing it to be visualized and analyzed rather than treated as a hidden byproduct of training. As a result, the prior provides structural completeness and plausibility during inference, particularly under challenging conditions such as heavy occlusion.

Upon completing the training, we obtain a model that enables accurate pose estimation, a categorical pose prior that encapsulates the general features of a category, and a hierarchical memory that stores

diverse prototypical poses for that category. We evaluate the effectiveness of our PPL across several human and animal pose estimation benchmarks. We visualize their pose priors to further interpret what our approach has learned. Additionally, we introduce an iterative inference strategy to estimate the poses of objects in occluded scenes using the trained hierarchical memory and the pose prior.

Our contributions are highlighted below:

**1.** We introduce the challenge of unsupervised categorical prior learning for pose estimation.

**2.** We propose a new method called Pose Prior Learner (PPL) for unsupervised pose estimation. PPL outperforms existing methods across several pose estimation benchmarks and offers explainable visualizations of pose priors. Notably, we found that predefined human priors are not always optimal. Our PPL even outperforms models using human-defined priors.

**3.** We introduce an explicit and symbolic pose prior that is represented in a structured form rather than being implicitly stored in model parameters. This enables the categorical structure to be directly interpreted, visualized, and examined, providing a clearer understanding of how the prior guides pose estimation.

**4.** During inference, we utilize an iterative strategy in which PPL progressively leverages priors to refine estimated poses by regressing them to the nearest prototypical poses stored in memory. Experimental results demonstrate that our PPL accurately estimates poses, even in occluded scenes.

## 2 RELATED WORKS

**Unsupervised Pose Estimation without Priors.** Numerous unsupervised learning methods without priors have been proposed to detect keypoints from images, which are then used to reconstruct images for supervision Li et al. (2021); Geng et al. (2021); Zhang et al. (2018); Sun et al. (2022); Thewlis et al. (2017); Jakab et al. (2020). For example, AutoLink He et al. (2022a) extracts keypoints from the image and estimates the strength of the links between pairs of keypoints. It then combines these keypoints with the link heatmap to reconstruct the randomly masked image. In these methods, keypoints are directly predicted from the image and supervised solely by image reconstruction, leading to potential detection of keypoints in background regions with complex textures. To alleviate this problem, BKind Sun et al. (2022; 2023) uses keypoints extracted from two video frames to reconstruct the pixel-level differences between these two frames. However, the lack of constraints on keypoint configuration and connectivity still undermines the reliability of their approach. In contrast, our PPL utilizes the learned pose prior as a constraint to mitigate these issues. There are also other category-level pose estimation methods Chen et al. (2020); Guo et al. (2022) which assume a canonical pose within a category and aim to estimate the pose for individual instances. Although these methods implicitly exploit structural regularities, the prior remains embedded in model parameters and is not explicitly extracted or analyzed. In contrast, we formulate categorical prior learning as a distinct problem: the goal is to explicitly distill a pose prior from raw data, while pose estimation is used only as a testbed to verify the learned prior. Our framework is fully unsupervised, requiring no pose annotations, CAD models, or external constraints, and it allows individual pose estimates to be summarized into a structured prior that gradually improves estimation accuracy during training. Besides, these methods output estimated individual poses only, without an explicit prior. In contrast, our model additionally outputs a learned categorical prior, which can be directly visualized and used to infer poses even under occlusion — a capability not explored in these previous works.

**Unsupervised Pose Estimation Incorporating Human-defined Priors.** Several methods utilize prior knowledge from human annotators to guide the pose estimation Chen & Dou (2021); Shi et al. (2023); Zhang et al. (2022b); Schmidtke et al. (2021); Yoo & Russakovsky (2023). Among these methods, Shape Template Transforming (STT) Schmidtke et al. (2021) applies affine transformations to a predefined pose prior, aligning it with the estimated pose from a video frame. By incorporating an additional frame from the same video to provide background information, an image reconstruction loss supervises the pose estimation process. The pose prior effectively guides pose estimation by constraining the shape of the human pose and the connectivity between body parts. However, pose priors are often difficult to obtain, requiring costly human annotations. Moreover, HPE Yoo & Russakovsky (2023) has shown that predefined pose priors are not always optimal, and tuning the shape of the prior can sometimes improve performance. Unlike these methods, our approach learns the prior directly from input images without any manual annotations, and models with our learned priors even outperform those using human-defined priors. Recently, Hedlin et al. (2024) leveraged a

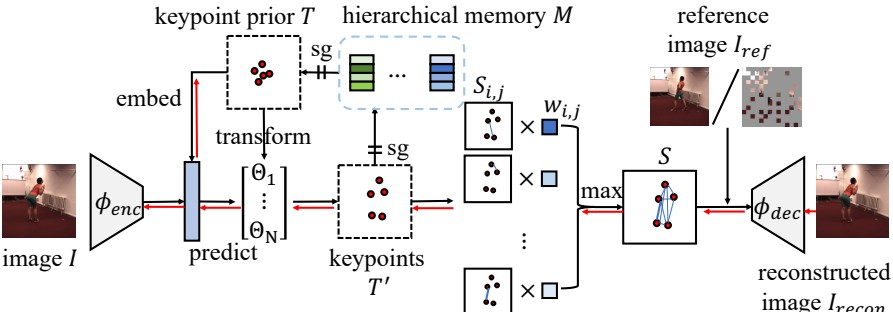

Figure 2: **Overview of our proposed Pose Prior Learner (PPL).** We first distill the keypoint prior from the hierarchical memory $M$. Features of the image $I$ and the embedding of the keypoint prior are concatenated to predict the affine transformation parameters. The keypoint prior is transformed and their pair-wise links are modulated with the connectivity prior $W$ to obtain the combined link heatmap $S$. The concatenation of the link heatmap $S$ and the reference image $I_{ref}$ is decoded to produce the reconstructed image $I_{recon}$. The $sg$ symbol represents the stopping gradient operation. The red arrows indicate the gradient flows during backpropagation based on image reconstruction. See **Section 3.4** for training details.

large pre-trained text-to-image Stable Diffusion model as its prior knowledge for pose estimation. In contrast, our model attains comparable pose estimation accuracy while training solely on the vision modality, with substantially reduced model size.

## 3    OUR PROPOSED POSE PRIOR LEARNER (PPL)

We introduce our proposed method, Pose Prior Learner (PPL). Given images featuring object instances from a specific category, PPL can accurately estimate the poses of the objects in that category while gradually learning a general pose prior through unsupervised learning. Note that our PPL requires no extra knowledge from human annotators. The architecture of PPL is presented in **Figure 2**.

Mathematically, we represent the topology for an object as a graph connecting keypoints with shared link weights, also known as connectivity. For a category of objects, its general pose prior $V$ is defined as $V = (T, W)$, where $T$ represents its keypoint prior and $W$ denotes the connectivity prior. Specifically, $T$ consists of $N$ keypoints: $T = [P_1, P_2, ..., P_N]$, where $P_i \in [-1, 1] \times [-1, 1]$ is the normalized 2D coordinates in the image pixel space. Unlike pre-defined priors, before training, $P_i$ does not explicitly encode the semantic parts of an object. $W$ is a $2D$ matrix of size $N \times N$, where each entry $w_{ij}$ in the matrix represents the connectivity probability between two keypoints $P_i$ and $P_j$. For instance, in the case of humans, the hand is connected to the torso via an arm; thus, the connectivity probability between these two parts should be higher than the connectivity probability between a hand and a foot. We initialize $W$ with random positive values. $T$ is decoded from a hierarchical memory $M$ storing prototypical poses, which is also randomly initialized.

The aim of PPL is to learn to correctly predict the image-specific keypoints $T'$ and their connectivity on input image $I$ from the general categorical pose prior $V = (T, W)$. Ideally, if PPL makes perfect predictions of the pose on $I$, by combining it with the background information, the reconstructed image $I_{recon}$ should match $I$ exactly. The background information is provided by the reference image $I_{ref}$, drawn from either image or video datasets. For video datasets, $I_{ref}$ corresponds to a randomly selected frame within the same video that depicts the same object in a different pose. For static image datasets, $I_{ref}$ is constructed by applying a random masking operation to the original image. Next, we introduce how to estimate $T'$ on $I$ using the keypoint prior $T$.

### 3.1    RECONSTRUCTION OF KEYPOINT CONFIGURATION WITH MEMORY

$M$ is a memory module that stores compositional representations of prototypical keypoint configurations. We organize $M$ hierarchically into $m$ memory banks $\{b_1, b_2, \ldots, b_m\}$, each containing $k$ learnable vectors of dimension $d$. This hierarchical design brings three advantages: (1) Hierarchical $M$ offers richer complexity, as its capacity can scale exponentially with the number of levels. (2) Organizing information across banks facilitates robust retrieval of relevant prototypes in uncertain or ambiguous cases (e.g., occlusion). When occlusion disrupts the feed-forward visual

pathway, the stored prototypical poses offer plausible hypotheses about the true pose given only partial observations. By iteratively refining its estimates against these prototypes, the network can "fill in" missing information and recover accurate poses, even under heavy occlusion. (3) Hierachical $M$ allows for efficient information retrieval at multiple levels: while $M$ encodes complete pose configurations, hirachical memory narrow down the search space by dividing $M$ into parts and individual memory banks capture finer sub-structures of poses. We provide a comparison with a single large memory bank in **Table A3** in **Appendix A8** to highlight the advantage of this hierarchical design.

**Encoding into Memory Space.** Given the estimated keypoints $T'$, we encode them into $m$ tokens of dimension $d$ using several MLP-Mixer blocks Tolstikhin et al. (2021), denoted as $MIX_{enc}$. The number of tokens equals the number of memory banks, with each token $g_i$ assigned to a distinct embedding space corresponding to memory bank $b_i$. Collectively, these tokens are $G = [g_1, g_2, \ldots, g_m] = MIX_{enc}(T')$.

**Memory Retrieval and Reconstruction.** If the vectors in $M$ learn to represent unique parts of prototypical poses, each $g_i$ should retrieve its most similar vector from bank $b_i$. We measure similarity using the $L_2$ distance, and denote the retrieved vector as $g'_i$. The retrieved set is $G' = [g'_1, g'_2, \ldots, g'_m]$. Finally, another series of MLP-Mixer blocks ($MIX_{dec}$) decodes $G'$ back into $N$ keypoints: $T'_{recon} = MIX_{dec}(G')$. See **Figure A6(a)** in **Appendix A7** for an illustration.

**Distilling the Keypoint Prior.** Unlike PCT Geng et al. (2023), where all memory vectors lie in a single embedding space, we assign each bank an independent space. This setup allows each bank to capture distinct pose components, which can then be distilled into a general pose prior. Specifically, let $MP(\cdot)$ denote mean pooling. PPL pools the $k$ vectors within each bank into one vector, resulting in $m$ pooled vectors. These are decoded into $N$ keypoints by $MIX_{dec}$: $T = MIX_{dec}([MP(b_1), MP(b_2), \ldots, MP(b_m)])$. See **Figure A6(b)** in **Appendix A7** for the schematic.

## 3.2 Transformation of the Keypoint Prior

As introduced above, given the keypoint prior $T$ distilled from the memory $M$, and the image $I$, we use a feature extractor $\phi_{enc}$ to extract its embedding $h_I$: $h_I = \phi_{enc}(I)$. $\phi_{enc}$ is a 2D-Convolution Neural Network (2D-CNN) trained from scratch. The keypoint prior $T$ is converted into an embedding $h_T$ via a series of fully connected layers. Together with $h_T$ as inputs, PPL learns to predict the affine transformation parameters $\Theta_i \in [\Theta_1, \Theta_2, ..., \Theta_N]$ for each keypoint $P_i$ in $T$ from $h_I$ via a two-layer fully connected network denoted as $FC(\cdot)$: $[\Theta_1, \Theta_2, \ldots, \Theta_N] = FC(h_I, h_T)$, where $\Theta_i = \left[ \left[ a^{(i)}, b^{(i)}, t_x^{(i)}; c^{(i)}, d^{(i)}, t_y^{(i)}; 0, 0, 1 \right] \right]$. $t_x^{(i)}$ and $t_y^{(i)}$ are the translations and $a^{(i)}, b^{(i)}, c^{(i)}, d^{(i)}$ are the coefficients that define rotation, scaling, and shear. Each point $P_i$ in $T$ can then be transformed by $\Theta_i$, resulting in keypoints $T'$ for image $I$: $T' = [P'_1, P'_2, ..., P'_N]$, where $[P'_i, 1]^\top = \Theta_i [P_i, 1]^\top$.

## 3.3 Connecting Keypoints Based on the Connectivity Prior

The connectivity of keypoints in objects is often fixed and rigid, for example, human arms maintain a relatively constant length, with a hand always connected to the torso via an arm and never connected to a foot. This rigidity in connectivity serves as a constraint, aiding in the regularization of pose estimation. In this section, we introduce the connectivity prior and explain how it can be used to regularize the connectivity strength between any pair of estimated keypoints in $T'$ on $I$.

Similar to AutoLink He et al. (2022a), PPL connects any two keypoints $P'_i$ and $P'_j$ in $T'$ to obtain differentiable link heatmap $S_{i,j} \in \mathbb{R}^{H \times W}$. Each 2D link heatmap represents a probability density map, where the pixel values along the link between two points are high, while other areas are assigned values close to zero. For any point $P'_i$, its strongest connectivity to any of the other points in $T'$ is activated on the combined link heatmap $S \in \mathbb{R}^{H \times W}$ via a max pooling operation over all the $N \times N$ link heatmaps: $S = \max_{i,j}^{N \times N}(w_{i,j} S_{i,j})$, where $w_{i,j}$ in the connectivity prior $W$ modulates the link heatmap $S_{i,j}$ based on whether the two keypoints $P'_i$ and $P'_j$ are physically connected. Ideally, if PPL correctly estimates the probability of physical links for an object category, $S_{i,j}$ will receive higher connectivity values, thereby activating the locations linking these two keypoints on the combined link map $S$.

Given the link map $S$ and the reference image $I_{ref}$, PPL can reconstruct the image $I$. $I_{ref}$ provides background information for reconstruction, while $S$ supplies foreground structural information by

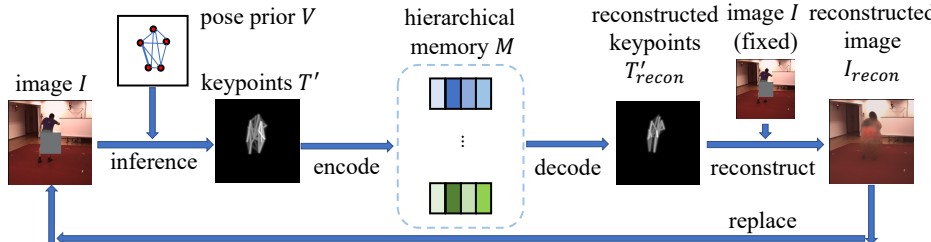

Figure 3: **Overview of the iterative inference strategy in our PPL (Section 3.4).** During inference, we iteratively use the reconstructed image $I_{recon}$ as input to estimate the pose $T'$. The hierarchical memory $M$ refines the estimated pose $T'$ and outputs $T'_{recon}$. The original image $I$ is used as the reference image to reconstruct the image $I_{recon}$. It is then used as the input image in the next iteration.

linking the estimated keypoints with the connectivity prior. Therefore, we concatenate $S$ and $I_{ref}$ and feed them into a 2D-CNN to perform the image reconstruction $I_{recon}$: $I_{recon} = \phi_{dec}(I_{ref}, S)$.

## 3.4 TRAINING AND INFERENCE

Our PPL is trained to jointly minimize all the four losses: the image reconstruction loss $L_{ir}$, the boundary loss $L_b$, the link regularization loss $L_l$, and the keypoint configuration reconstruction loss $L_{kr}$. We describe these losses in detail below and present their ablation results in **Table A3** in **Appendix A8**. Additionally, we use three training techniques for stable convergence of the network. We present the details of these training techniques in **Appendix A2**.

**Image Reconstruction Loss.** If PPL correctly estimates the pose of image $I$, the reconstructed image $I_{recon}$, based on the estimated pose, should be identical to $I$. Here, we adopt the semantic consistency between $I$ and $I_{recon}$, rather than strict pixel-wise correspondence for the reconstruction. Therefore, ensuring the quality of $I_{recon}$ encourages PPL to improve its pose estimation accuracy. To achieve this, we apply a perceptual loss on the embeddings of $I$ and $I_{recon}$, extracted using a frozen feature extractor $\psi(\cdot)$ from the VGG19 network pre-trained on ImageNet Russakovsky et al. (2015). The perceptual loss is defined as: $L_{ir} = \|\psi(I_{recon}) - \psi(I)\|_1$. We further evaluate the influence of the perceptual loss by replacing it with the MSE loss at the pixel level in the model analysis ( **Table A3** of **Appendix A8**). Results show that pixel-level reconstruction is less effective than perceptual loss.

**Boundary Loss.** To prevent the keypoint prior from being transformed outside the image boundary, we limit the transformed keypoints to be within the image with a regularization loss: $L_b = \sum_{*\in\{x,y\}}(|P'_{i,*}|$ if $|P'_{i,*}| > 1$, else 0), where $P'_{i,x}$ and $P'_{i,y}$ are the normalized $x$ and $y$ coordinate of the keypoint $P'_i$ respectively. Empirically, we found that this loss encourages PPL to produce more accurate pose predictions and helps the model converge faster.

**Link Regularization Loss.** A person's arm always maintains a fixed length regardless of the poses. Thus, we propose the constraint that links should be assigned a high weight if they do not vary significantly in length before and after the transformation. Since limb rigidity is inherently defined in 3D space, we introduce it as an auxiliary constraint in 2D that approximates this property, primarily to accelerate convergence and stabilize training. The loss $L_l$ encourages the preservation of the link lengths during pose estimation. It is defined as: $L_l = \sum_{i,j} w_{i,j}\|l(P_i, P_j) - l(P'_i, P'_j)\|_1$, where $l(\cdot)$ is the $L2$ distance between two keypoints before and after the transformation.

**Reconstruction Loss on Keypoint Configurations.** In **Section 3.1**, given a collection of token representations in $G$, PPL retrieves the most similar vectors from each memory bank of the hierarchical memory $M$ and generates $G'$ in a non-differentiable manner. To ensure that $M$ learns to store meaningful token embeddings that represent compositional parts of poses, the retrieved vectors $G'$ from $M$ should closely match $G$. Moreover, if these compositional parts are structured correctly, the vectors should be able to decode into meaningful keypoint configurations $T'_{recon}$ that are close to the original keypoint configurations $T'$. Therefore, we introduce the keypoint configuration reconstruction loss defined as: $L_{kr} = \|T'_{recon} - T'\|_2 + \|G - G'\|_2$.

**Iterative Inference.** The memory in PPL stores learned prototypical poses, making it well-suited for correcting inaccurate pose estimates—especially under occlusion—through an iterative autoregressive process during inference (**Figure 3**). Iterative inference leverages this memory to progressively refine

Table 1: **Keypoint detection on the video datasets Human3.6m and Taichi, and the image dataset CUB-200-2011.** For the video datasets Human3.6m and Taichi, we use video frames as $I_{ref}$ during training. For all the CUB evaluation subsets, we instead use randomly masked images as $I_{ref}$. We report mean $L_2$ error ($\downarrow$) normalized by image resolution for Human3.6m and CUB-200-2011, and summed $L_2$ error ($\downarrow$) for Taichi. The smaller the error, the better the model performance is. We use resolution $256 \times 256$ for Taichi and $128 \times 128$ for CUB-200-2011. For Human3.6m, we report results under both resolutions. Following He et al. (2022a), we use different numbers of keypoints for the evaluation subsets of CUB-200-2011: 10 for CUB-Aligned and 4 for CUB-001, CUB-002, CUB-003, and CUB-All. The best results are highlighted in bold. A dash (–) indicates that the result of the corresponding method is not applicable in the literature.

| Method | Human3.6m | | Taichi | CUB-Aligned | CUB-001 | CUB-002 | CUB-003 | CUB-All |
|---|---|---|---|---|---|---|---|---|
| | Res. 128 | Res. 256 | | | | | | |
| Jakab Jakab et al. (2020) | - | 2.73 | - | - | - | - | - | - |
| Thewlis Thewlis et al. (2017) | - | 7.51 | - | - | - | - | - | - |
| Lorenz Lorenz et al. (2019) | - | 2.79 | - | - | - | - | - | - |
| Schmidtke Schmidtke et al. (2021) | - | 3.31 | - | - | - | - | - | - |
| BKind Sun et al. (2022) | 2.44 | - | - | - | - | - | - | - |
| Siarohin Siarohin et al. (2021) | - | - | 389.78 | - | - | - | - | - |
| LatentKeypointGAN He et al. (2021) | - | - | 437.69 | 5.21 | 22.6 | 29.1 | 21.2 | 14.7 |
| GANSeg He et al. (2022b) | - | - | 417.17 | 3.23 | 22.1 | 22.3 | 21.5 | 12.1 |
| Zhang Zhang et al. (2018) | - | 4.91 | 343.67 | 5.36 | 26.9 | 27.6 | 27.1 | 22.4 |
| AutoLink He et al. (2022a) | 2.76 | - | 316.10 | 3.51 | 20.2 | 19.2 | 18.5 | 11.3 |
| PPL (ours) | **1.92** | **2.56** | **293.35** | **3.19** | **19.3** | **18.6** | **17.3** | **10.5** |

predictions alongside image reconstruction. In every iteration, we take the reconstructed image $I_{recon}$ from the last iteration (the original image $I$ for iteration 0) as the input. We infer its keypoints $T'$ as the output keypoints of the current iteration. The hierarchical memory $M$ is used to reconstruct $T'$ and the reconstructed keypoints $T'_{recon}$ are used to obtain the reconstructed image $I_{recon}$. $I_{recon}$ is then used as the input for the next iteration. We keep the original occluded image $I$ as the reference image for all the iterations. 4 iterations are used for every experiment.

In our experiments (see **Section 4.3**), we demonstrate that iterative inference outperforms one-step correction. This advantage stems from several key factors: (1) A single correction often fails to capture complex dependencies and may introduce additional errors. In contrast, autoregressive inference progressively refines predictions, enhancing both accuracy and consistency. (2) For tasks like pose estimation, initial predictions are often ambiguous. Iterative updates enable the model to integrate contextual cues at each step, thereby resolving uncertainties more effectively. (3) Iterative inference also encourages the model to reason about structured dependencies across spatial and temporal domains, improving robustness to scene variation, object occlusion, and background noise.

**Training and Implementation Details.** We use the Adam optimizer with a learning rate of $10^{-3}$, training for 50 epochs. The learning rate for link weights is scaled by 512 to address the small gradients of SoftPlus near zero. We conduct experiments using the link thickness $\sigma^2 = 5 \times 10^{-4}$ across all benchmark datasets, where we adopt the same definition of $\sigma$ used in He et al. (2022a). We use 34 memory banks, each of which contains 16 vectors of dimension 512. All results are averaged over 3 experiment runs. We train and test PPL on a single NVIDIA RTX A6000 GPU with the batch size of 64. Compute efficiency is reported in **Table A1**. Details are in **Appendix A1**.

## 4 EXPERIMENTS

For quantitative experiments in pose estimation, we use three public datasets: Human3.6m Ionescu et al. (2013), Taichi Siarohin et al. (2019b;a), and CUB-200-2011 Wah et al. (2011) We employ summed or normalized L2 distance between predicted and ground-truth keypoints as the evaluation metric for pose estimation tasks. We set the number of keypoints $N$ to be the same as He et al. (2022a). We use PPL trained on Human3.6m for all ablation studies and experiments in occluded scenes. For qualitative visualizations of the learned pose priors, we additionally include Youtube dog videos, Flowers Nilsback & Zisserman (2008), 11k-Hands Afifi (2019), and Horses Zhu et al. (2017). To train PPL, we utilize video frames as $I_{ref}$ on the Human3.6m, Taichi, and YouTube dog videos, while randomly masked images are used as $I_{ref}$ for other datasets. See **Appendix A3** for dataset details. Besides, we conduct additional image recognition experiments using PPL to demonstrate that

the learned pose prior is transferable and can be effectively applied to other downstream tasks beyond pose estimation. The results and experimental details are in **Appendix A10**.

## 4.1 Unsupervised Categorical Pose Estimation

We compare the keypoint detection results of PPL with other unsupervised pose estimation methods and present the results in **Table 1**. On all datasets, PPL outperforms all baselines across all image resolutions. We note that STT Schmidtke et al. (2021), a baseline with human-defined priors, underperforms PPL. Consistent with HPE Yoo & Russakovsky (2023), this suggests that pre-defined priors are not always optimal. PPL can learn more representative priors that outperform those manually defined. Among the baselines, AutoLink He et al. (2022a) also incorporates learnable connectivity priors; but its performance is inferior to PPL due to the absence of hierarchical memory.

To demonstrate that PPL's advantage comes from the learned priors rather than temporal consistency in video datasets, we treat all video frames as independent images and use randomly masked images as $I_{ref}$ to train PPL on the video datasets. We denote the resulting model as PPL*. Notably the results of PPL* on Human3.6m and Taichi in **Table A4** in **Appendix A8** show that even without temporal consistency in video reference frames as supervision, PPL* continues to outperform AutoLink. This highlights that the effectiveness of PPL stems primarily from its pose prior, rather than from cross-frame reconstruction.

Additionally, we provide the result comparisons between our PPL and Hedlin et al. (2024) in **Table A2** in **Appendix A4**. It is noteworthy that PPL performs competitively with Hedlin et al. (2024), which is based on priors from pre-trained text-image stable diffusion models, despite PPL being much smaller in size and relying solely on the visual modality rather than both text and vision.

We also present visual comparisons of the estimated poses between our PPL and AutoLink in **Figure 4(c)**. From scenes without complex backgrounds (Row 1), both AutoLink and PPL can predict correct poses. However, in more challenging settings—such as those with high-contrast backgrounds (Row 2) or where arms blend into the background (Row 3)—AutoLink occasionally misplaces keypoints on the background, resulting in implausible poses. In contrast, PPL consistently attends to the human body, leveraging its learned pose prior to enforce realistic body constraints.

**Visualization of Pose Estimation.** We provide the visualization results of pose estimation on Human3.6m (**Figure A3(a)**), YouTube dog videos (**Figure A3(b)**), 11k-Hands (**Figure A3(c)**), CUB-200-2011 (**Figure A3(d)**), Horse (**Figure A3(e)**), and Flowers (**Figure A3(f)**) in **Figure A3** in **Appendix A5**. The visualization results demonstrate that PPL can learn categorical priors and estimate poses for various object categories, without any external annotations. For example, in Row 2, Column 2 of **Figure A3(a)**, PPL correctly estimates the bowing pose of a person. In Row 2, Column 5 of **Figure A3(b)**, PPL correctly estimates the pose of a dog lowering down its head. Moreover, we found that the quality of estimated dog poses is lower than that of humans, primarily because dogs exhibit much greater variability in pose, breed, size, and fur texture. Additional visualization results for Flowers, Hands, and Horses in **Figure A3** in **Appendix A5** further demonstrate that our PPL can be applied for various categories. Notably, flowers are rigid objects with no degrees of freedom like animals. Yet, our PPL still learns meaningful categorical priors for them. We also present visualizations of semantic consistency in predicted keypoints in **Figure A8** of **Appendix A9**, which further demonstrates the capability of PPL in achieving semantic alignment.

**Visualization of the Pose Prior Changing with the Training Epochs.** We visualize the progressively learnt pose priors by our PPL as a function of training epochs. **Figure 4(b)** illustrates that the keypoint prior converges to a human shape by the early stage of training (epoch 5). Notably, the learnable keypoints align with the human joints defined in the literature, and the connectivity among keypoints corresponds to the physical connections between body parts. As training continues, the connectivity prior gradually learns the skeletal structure of the human body, with irrelevant links between keypoints diminishing over time, as seen when comparing epochs 15 and 20. Note that we do not visualize the learned prototypical poses stored in PPL's memory. This is because our model does not rely on a single, flat global memory. Instead, it uses multiple hierarchical memory banks, each encoding partial structural information at different levels of abstraction. Consequently, there is no one-to-one correspondence between any single "prototype" and a complete pose template.

Table 2: **Keypoint detection results of our PPL variants on the Human3.6m dataset.** All results in mean $L2$ errors ($\downarrow$) are normalized by the image resolution of $256 \times 256$. Both keypoint prior (Row 1-2) and Connectivity prior (Row 3-4) can be either pre-defined (Pre.) or randomly initialized (Rand.). During training, the parameters in both the priors can be either frozen ($\boldsymbol{\times}$) or learnable ($\checkmark$). The last column (From Mem) shows the result of our default PPL method. Its keypoint prior is initialized from memory (From Mem). Its connectivity prior is randomly initialized (Rand.) and learnable ($\checkmark$) during training. Best is in bold.

| | | 1 | 2 | 3 | 4 | 5 | 6 | 7 | 8 | 9 | 10 | 11 |
|---|---|---|---|---|---|---|---|---|---|---|---|---|
| Keypoint prior | Initialization | Pre | Pre | Pre | Pre | Pre | Pre | Rand | Rand | Rand | Rand | From mem |
| | Trainable | $\checkmark$ | $\checkmark$ | $\boldsymbol{\times}$ | $\boldsymbol{\times}$ | $\checkmark$ | $\boldsymbol{\times}$ | $\checkmark$ | $\boldsymbol{\times}$ | $\checkmark$ | $\boldsymbol{\times}$ | $\boldsymbol{\times}$ |
| Connectivity prior | Initialization | Pre | Pre | Pre | Pre | Rand | Rand | Pre | Pre | Rand | Rand | Rand |
| | Trainable | $\checkmark$ | $\boldsymbol{\times}$ | $\checkmark$ | $\boldsymbol{\times}$ | $\checkmark$ | $\checkmark$ | $\checkmark$ | $\checkmark$ | $\checkmark$ | $\checkmark$ | $\checkmark$ |
| Normalized $L2$ Error | | **2.51** | 2.66 | 2.58 | 2.70 | 2.54 | 2.61 | 2.68 | 2.72 | 2.75 | 2.83 | 2.56 |

## 4.2 Ablation Studies

**Ablation on Prior Variants.** We investigate how different initializations of connectivity and keypoint priors affect pose estimation and whether further refining these priors enhances performance. From **Table 2**, we obtain several key insights: (1) Models with frozen, human-defined priors (Column 4) perform worse than PPL, indicating that PPL learns more representative priors than those predefined by humans. (2) Refining pre-defined keypoint and connectivity priors (Column 1) outperforms our default PPL, suggesting that PPL can enhance models with human-defined priors through refinement. (3) Interestingly, randomly initializing either keypoint or connectivity priors, followed by refinement during training (Columns 5-9), yields comparable performance to models with human-defined priors. This suggests that human-defined priors may not be necessary for effective pose estimation. (4) Surprisingly, freezing randomly initialized keypoint priors also results in reasonable pose estimation accuracy, though it is still lower than PPL's default performance (Columns 7 and 9). (5) In contrast to (4), freezing random connectivity priors prevents the model from converging, implying that connectivity priors play a more critical role in guiding pose estimations than keypoint priors.

**Ablation on Number & Dimension of Vectors in Each Memory Bank.** In our hierarchical memory, we used 34 memory banks. Here, we analyze the impact of the number of vectors per memory bank and the dimension of each vector on PPL's pose estimation performance. From **Figure A7** in **Appendix A8**, we observed that PPL remains robust across different vector counts and dimensions, although performance slightly improves with more vectors of higher dimensions. As a result, we fixed 16 vectors per memory bank, each with a dimension of 512, for all experiments.

**Ablation on Number of Keypoints.** We varied the number of keypoints in the pose priors from 4 to 32. The results in **Figure A7** in **Appendix A8** show that pose estimation accuracy improves as the number of keypoints in the prior increases. However, using 32 keypoints offers limited improvement compared to PPL with 16 keypoints.

## 4.3 Using Learned Priors for Inference in Occluded Scenes

To verify the robustness of PPL in pose estimation in occluded scenes, we divide the image into $32 \times 32$ patches and apply two masking techniques: RandomMasking and CenterMasking. In RandomMasking, we randomly mask a certain proportion of image patches, with the proportion ranging from 0.1 to 0.4. In CenterMasking, we mask only the center region of the image, gradually increasing the masking size from $4 \times 4$ to $12 \times 12$ patches. We explore the effect of occluded areas on PPL. From **Figure A4** and **Figure A5** in **Appendix A6**, we observe that at iteration 0, as the occluded areas increase, overall performance declines with larger occlusions. However, with our iterative inference strategy, PPL effectively infers the missing parts of the poses by utilizing prototypical poses stored in hierarchical memory and the learned priors. Notably, it restores partially occluded poses to reasonably complete full-body poses, leading to a lower L2 error, comparable to those without occlusion. This effect is more pronounced with smaller occluded areas.

**Visualization of Pose Estimation with Occlusion.** We present the estimated poses by our PPL for occluded images as a function of the number of inference iterations in **Figure 4(a)**. Across both RandomMasking and CenterMasking, with our iterative inference strategy, PPL successfully reconstructs the occluded image parts after three iterations and meanwhile, predicts reasonable

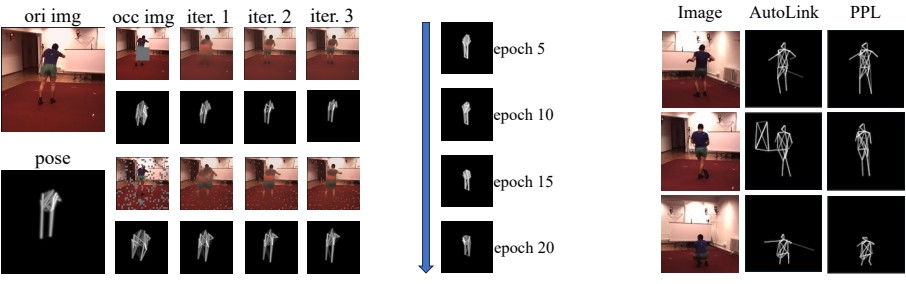

(a) Occluded Human Pose Estimation     (b) Prior Update     (c) Comparison with AutoLink

Figure 4: **Visualization results of poses estimation on Human3.6m.** (a) Pose estimation on occluded images in Human3.6m. The first column shows the original image and its estimated pose by PPL. Columns 2-5 show the iterative inference process where the reconstructed images by PPL (Row 1 and 3) are fed back to itself for estimating poses (Rows 2 and 4) on occluded images either using CenterMasking (Row 1 and 2) or RandomMasking (Row 3 and 4). (b) The pose prior evolves as a function of training epochs (from top to bottom). (c) Comparison between PPL and AutoLink He et al. (2022a) in estimated poses on example images from Human3.6m. The left column shows testing images, the middle column and the right column show estimated poses by AutoLink and PPL respectively. All results are obtained with 16 keypoints.

full-body poses. We include additional qualitative samples on occlusions in **Figure A1** in **Appendix A5**, showing that PPL consistently outperforms AutoLink by reconstructing plausible poses even with partial visual information. For example, our method consistently predicts legs with reasonable length, whereas AutoLink fails to do so. This emergent consistency suggests that our model captures meaningful structural cues, indicating the correctness and reliability of our method.Generally, incorrect predictions are corrected by the memory, particularly under small-area occlusions. However, severe occlusions can lead to plausible yet incorrect reconstructions, potentially compounding errors. A failure example is provided in **Figure A2** in **Appendix A5** where PPL fails to reconstruct the ground truth pose with large occlusion.

**Prior Learning Extends Beyond Pose Estimation.** We emphasize that pose estimation serves only as a testbed for evaluating the learned priors, which are expected to generalize to other tasks such as visual recognition. To examine this, we evaluate the prior-learning mechanism on 6-way image classification on Yoga82-Level1 Verma et al. (2020) and 10-way image classification on CIFAR-10 Krizhevsky et al. (2009), using ResNet-50 He et al. (2016) with and without PPL (implementation details in **Appendix A10**). As shown in **Table A5**, the same PPL module can be integrated into the ResNet-50 backbone without modifying the classification objective, and it consistently improves top-1 classification accuracy under occlusion. These results support the view that explicit prior learning is a general mechanism that extends beyond pose estimation.

## 5 DISCUSSION

We introduce the challenge of unsupervised categorical prior learning and highlight its significance in pose estimation. To address this, we propose a novel method called Pose Prior Learner (PPL). PPL utilizes a hierarchical memory to store compositional parts of learnable prototypical poses, which are distilled into a general pose prior for any object category. Our experimental results show that PPL requires no additional human annotations and outperforms recent competitive baselines in pose estimation. Notably, the learned prior proves to be even more effective in pose estimation than methods that rely on human-defined priors. With hierarchical memory and learned priors, PPL can perform iterative inferences and robustly estimate poses in occluded scenes. More importantly, this work proposes a new perspective on prior learning: we first estimate individual instances, and then distill their common structure into a categorical prior that captures the shared configuration of the category. This shows that prior knowledge can naturally emerge purely from visual observations.

Despite strong results of PPL, our work has several limitations. Currently, it learns only 2D priors, which limits its ability to model 3D rotations and significant shape variations. Future work will focus on extending PPL to 3D priors, and integrating more powerful backbones, such as Vision Transformers, to capture richer and more accurate priors.

ACKNOWLEDGEMENTS

This research is supported by the National Research Foundation, Singapore under its NRFF award NRF-NRFF15-2023-0001 and Mengmi Zhang's Startup Grant from Nanyang Technological University, Singapore.

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

## A1 IMPLEMENTATION DETAILS OF OUR POSE PRIOR LEARNER

We use the Adam optimizer with a learning rate of $10^{-3}$ and a batch size of 64, training for 50 epochs. Unless specified, all images are resized to $256 \times 256$. The learning rate for link weights is scaled by 512 to address the small gradients of SoftPlus near zero. We conduct experiments using the link thickness $\sigma^2 = 5 \times 10^{-4}$ across all benchmark datasets, where we adopt the same definition of $\sigma$ used in He et al. (2022a). For the hierarchical memory, we use 34 memory banks, each of which contains 16 vectors of dimension 512, for all experiments. We utilize video frames as $I_{ref}$ for the Human3.6m, Taichi, and YouTube dog videos, while randomly masked images are used as $I_{ref}$ for other datasets. On Human3.6m and CUB-200-2011, we report the results in the mean $L2$ error between the predicted keypoints and the ground truth, normalized by the image size. For Taichi He et al. (2022a); Siarohin et al. (2021); He et al. (2021; 2022b); Zhang et al. (2018), we use the summed $L2$ error computed at a resolution of $256 \times 256$. We excluded the DeepFashion Liu et al. (2016), CelebA Liu et al. (2015), and Zebra Zhu et al. (2017) datasets used by AutoLink because they either lack structural or background complexity, offer limited pose variation, or are redundant with existing categories. All results are averaged over 3 experiment runs. We train and test PPL on a single NVIDIA RTX A6000 GPU with the batch size of 64. The training and inference efficiency are presented in **Table A1**. Regarding computational efficiency, our hierarchical memory and prior extraction modules are lightweight with about 2.4M parameters only. This is compared against most of the large diffusion models which use pre-trained large language model as priors (usually over 900M parameters).

Table A1: **Training and inference efficiency of PPL on a single NVIDIA RTX A6000 GPU with batch size of 64.**

| Resolution | Training Time (per batch) | Inference Speed (0 iteration, img/s) | Inference Speed (3 iterations, img/s) | GPU Memory (usage) |
|---|---|---|---|---|
| $128 \times 128$ | $\sim$1.3 sec | $\sim$145 | $\sim$64 | $\sim$9.2 GB |
| $256 \times 256$ | $\sim$4.0 sec | $\sim$67 | $\sim$24 | $\sim$37.0 GB |

## A2 TRAINING TECHNIQUES OF OUR POSE PRIOR LEARNER

To ensure convergence and stability during the training of PPL, we introduce three gradient dettachment techniques: (1) To address the broken gradient issue during the quantization step from $G$ to $G'$, we adopt the approach from VQ-VAE Van Den Oord et al. (2017). Specifically, our PPL copies the gradients of $G'$ to $G$ for backward propagation, allowing the gradients to flow through the quantization step. (2) The hierarchical memory $M$ is updated using an exponential moving average to smooth the gradient updates, particularly during the early stages of training when $G$ can be quite noisy. This approach helps stabilize the learning process and ensures that $M$ retains more reliable information over time. (3) For $M$ to learn effective representations of $G$, it requires an accurate estimation of $T'$, which depends on a good prior $V$ distilled from $M$. This creates a chicken-and-egg problem that complicates training. To address this, we introduce two gradient detachments to separate the training processes. First, we detach the gradients from $T$ and train the keypoint transformation and image reconstruction pathway, as shown by the red arrows in **Figure 2**. Second, we detach the gradients from $T'$ to train the memory encoder and decoder, $MIX_{\text{enc}}$ and $MIX_{\text{dec}}$, as indicated by the green arrows in **Figure A6(a)** in **Appendix A7**.

## A3 DATASETS

**Human3.6m Ionescu et al. (2013)** is a standard benchmark dataset for human pose estimation, consisting of 3.6 million video frames. These frames include both 3D and 2D keypoints and were captured in a controlled studio environment with a static background, featuring various actors. We adhere to the approach outlined in Zhang et al. (2018); He et al. (2022a), focusing on six activities: direction, discussion, posing, waiting, greeting, and walking. We use subjects 1, 5, 6, 7, 8, and 9 for training; and we use subject 11 for testing.

**Taichi Siarohin et al. (2019b;a)** consists of 3,049 training videos and 285 test videos performing Tai-Chi, with diverse foreground and background appearances. Following the approach in Siarohin et al. (2021), we use 5,000 frames for training a linear regression and 300 frames for testing.

**YouTube dog videos** are videos with green backgrounds collected from YouTube to further qualitatively demonstrate the performance of PPL. Existing dog datasets are not suitable, as they often contain multiple, partially occluded dog instances. Therefore, we curated a custom dataset using 20 YouTube videos with a sampling frequency of 30 frames per second. We manually curate 2,000 training frames by selecting those frames with clear dog poses and removing redundant or similar ones. All videos in our custom dataset are publicly available, and are used strictly for research purposes in compliance with applicable legal and ethical guidelines. This dataset allows us to demonstrate PPL's ability to learn pose priors for non-human categories without using ground truth poses. All images are trained and tested at a resolution of $256 \times 256$. We use 10 keypoints for this category and provide the visualization of the estimated poses on the test videos. The dataset will be publicly released together with other data, models, and source code.

**CUB-200-2011 Wah et al. (2011)** contains 11788 images of birds across 200 bird species with detailed annotations such as bounding boxes, part locations, and attributes. We crop and align the images according to He et al. (2022a). We use the train/val/test split of Choudhury et al. (2021). The dataset supports both aligned and non-aligned evaluation protocols, where aligned evaluation uses cropped bird regions while non-aligned evaluation relies on the original full images, making the task more challenging. The aligned subset CUB-Aligned and non-aligned subsets CUB-001/002/003/all are used for evaluation. All images are trained and tested at a resolution of $128 \times 128$.

**11k-Hands Afifi (2019), Horse Zhu et al. (2017), and Flowers Nilsback & Zisserman (2008)** are used for qualitative visualization. Images with multiple horses are removed and all horses are aligned to face left. All images are trained and tested at a resolution of $128 \times 128$.

## A4 ADDITIONAL COMPARISON WITH METHODS USING MULTI-MODAL KNOWLEDGE

To provide broader context, we further compare our method with Hedlin et al. (2024). Hedlin et al. (2024) requires a Conditional Stable Diffusion Model pre-trained with large-scale multimodal data (image and text) to start with, thereby leveraging knowledge from an additional modality (text). The comparison results are presented in **Table A2**. Compared with Hedlin et al. (2024), our PPL method outperforms this baseline on Human3.6m and achieves comparable results on Tai-Chi for human pose estimation. For bird pose estimation, PPL achieves stronger performance on the CUB aligned evaluation set but lags behind on the non-aligned set. Overall, PPL employs a substantially smaller model, yet remains competitive with multimodal methods trained with extensive language supervision.

Table A2: Comparison with Hedlin et al. (2024) on keypoint detection. For Human3.6m and Taichi, we report the mean L2 error normalized by the image resolution of 128×128 and the summed L2 error at a resolution of 256×256, respectively. For CUB-200-2011, we report the mean L2 error normalized by the resolution of 128×128. We use different numbers of keypoints for the subsets of CUB-200-2011: 10 for CUB-Aligned and 4 for CUB-001, CUB-002, CUB-003, and CUB-All. The best results are in **bold**.

|  | Human3.6m | Taichi | CUB-Aligned | CUB-001 | CUB-002 | CUB-003 | CUB-all |
|---|---|---|---|---|---|---|---|
| Hedlin et al. (2024) | 4.45 | **234.89** | 5.06 | **10.5** | **11.1** | **10.3** | **5.4** |
| PPL (ours) | **1.92** | 293.35 | **3.19** | 19.3 | 18.6 | 17.3 | 10.5 |

## A5   ADDITIONAL VISUALIZATION RESULTS OF POSES UNDER OCCLUSION AND PRIORS FROM OTHER CATEGORIES

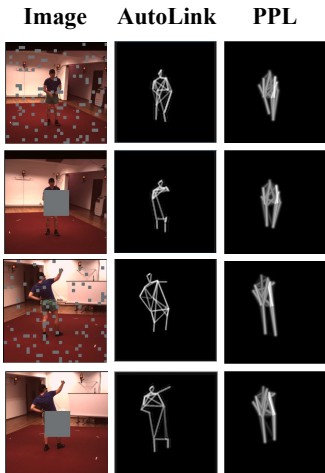

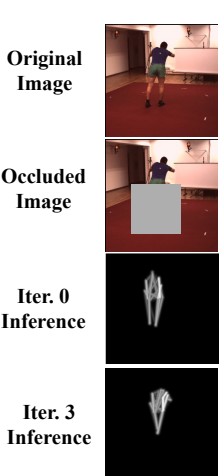

Figure A1: **Comparison between PPL and AutoLink pose estimation results on occluded images on Human3.6m.** The left column shows testing images, the mid column shows results of AutoLink, and the right column shows results from PPL. All results are obtained with 16 keypoints.

Figure A2: **One example where PPL fails to estimate the right pose when there is a large occlusion.** From left to right, we show the original unoccluded image, the largely occluded image, the result without iterative inference (iteration 0), and the final result with iterative inference (iteration 3).

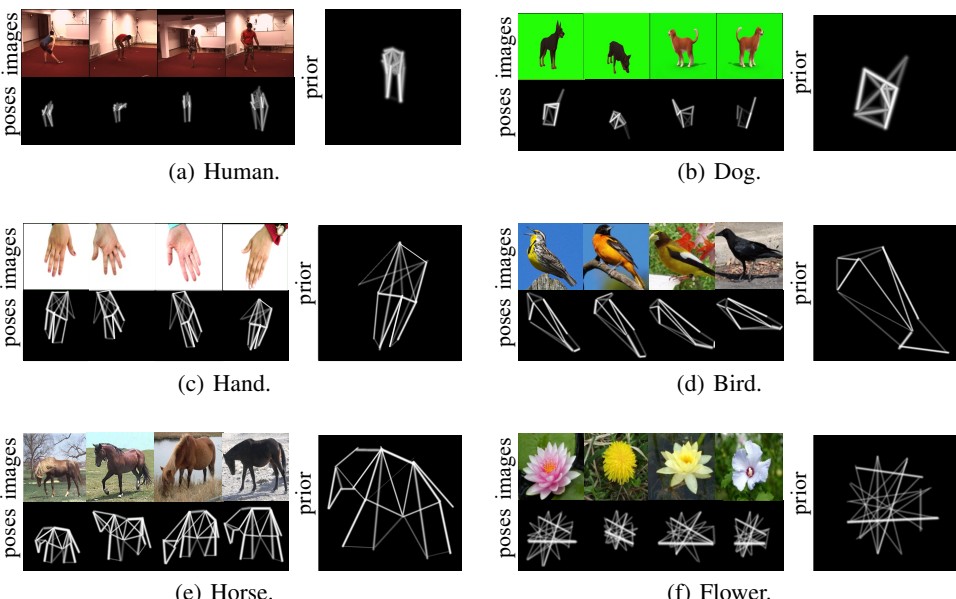

Figure A3: **Additional visualization on (a) Human3.6m, (b) Youtube dog videos, (c) 11k-Hands, (d) CUB-200-211, (e) Horse, and (f) Flowers.** Columns 1-4 of every subfigure show the original images and corresponding pose estimation results by PPL. The column 5 of every subfigure shows the learned prior for the category.

## A6 PPL RESULTS OF KEYPOINT DETECTION ON OCCLUDED IMAGES

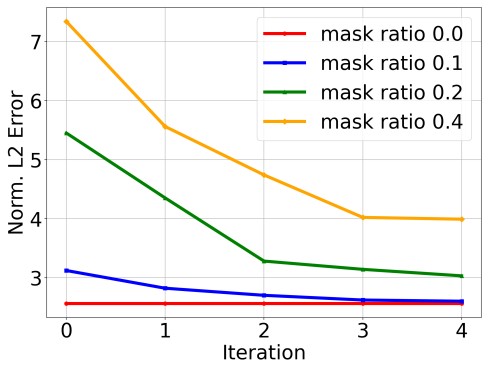

Figure A4: **PPL results of keypoint detection as a function of number of inference iterations on images with RandomMasking from Human3.6m.** The "mask ratio" in the legend specifies the masked proportion on the $32 \times 32 = 1024$ image patches.

Figure A5: **PPL results of keypoint detection as a function of number of inference iterations on images with CenterMasking from Human3.6m.** The "mask size" in the legend refers to the width and height of the masked region, on 1024 image patches.

## A7 RETRIEVAL AND DISTILLATION OF THE PROPOSED HIERARCHICAL MEMORY IN PPL

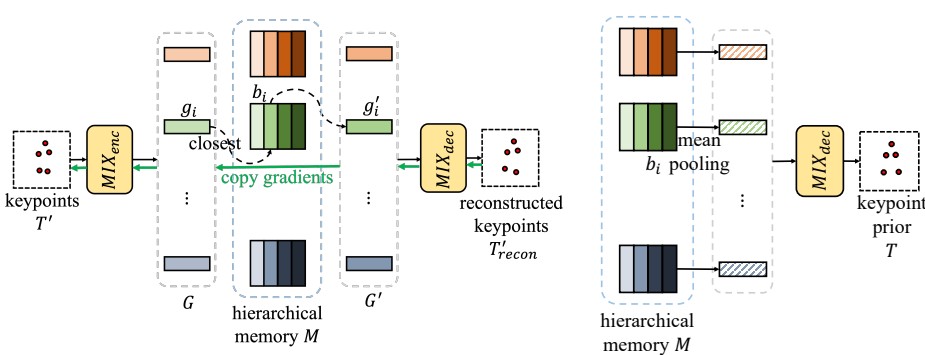

(a) Keypoint configuration reconstruction.

(b) Memory distillation.

Figure A6: **Retrieval and distillation of the proposed hierarchical memory in our PPL.** (a) The hierarchical memory $M$ is trained to reconstruct the keypoints $T'_{recon}$. $T'$ is encoded into $m$ tokens by the MLP-Mixer blocks $MIX_{enc}$. Each token $g_i$ retrieves its closest vector $g'_i$ in memory bank $b_i$. The resulting $m$ vectors are decoded by the MLP-Mixer $MIX_{dec}$ into the reconstructed keypoints $T'_{recon}$. The green arrows indicate the gradient flows during backpropagation based on the reconstruction of keypoint configurations. See **Section 3.4** for training details. (b) The hierarchical memory $M$ is distilled into the keypoint prior $T$. Vectors in every memory bank $b_i$ are mean-pooled into one vector, and the resulting $m$ vectors are decoded by $MIX_{dec}$ into the keypoint prior $T$. See **Section 3.1** for details.

## A8 ADDITIONAL ABLATION RESULTS

**Number of memory bank vectors and keypoints in Human3.6m.** We vary the dimensionality of the vectors in each memory bank to examine its impact on pose estimation accuracy. In addition, we analyze how the number of keypoint priors influences performance. Results are shown in **Figure A7**.

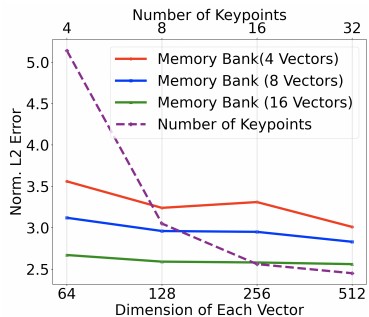

Figure A7: **Ablation of our PPL method on memory bank vectors and number of keypoints in Human3.6m.** The upper horizontal axis is the number of keypoints (ranging from 4 to 32) and the lower horizontal axis is the dimension of memory bank vectors (ranging from 64 to 512). The dashed purple line is for ablations on number of keypoints and the solid lines are for ablations on memory bank vectors.

**Ablations on loss components.** Our method employs four loss components: Image Reconstruction Loss, Boundary Loss, Link Regularization Loss, and Keypoint Configuration Reconstruction Loss. The Image Reconstruction Loss provides the primary training signal, ensuring that the learned keypoints are semantically meaningful and spatially aligned with the object. The Keypoint Configuration Reconstruction Loss is critical for the successful training of the hierarchical memory. The Boundary Loss and the Link Regularization Loss are auxiliary constraints that mainly stabilize and accelerate convergence. Without the Boundary Loss, the network becomes unstable during early training. Replacing it with a sigmoid function causes a clear performance drop, increasing the normalized L2 error from 2.56 to 2.70 on Human3.6M at resolution 256×256. Similarly, removing the Link Regularization Loss leads to a measurable decline, with the error rising from 2.56 to 2.64. When the perceptual loss is replaced with MSE, the normalized L2 error increases to 2.84, indicating a notable drop in performance. The comparison results are shown in **Table A3**.

Table A3: **Ablation studies on Human3.6M at resolution** $256 \times 256$. Numbers indicate normalized L2 error ($\downarrow$, lower is better). Best is in bold.

| Ablation Results | Normalized L2 Error |
|---|---|
| PPL-full (ours) | **2.56** |
| w/o Boundary Loss | training unstable |
| Boundary Loss $\rightarrow$ Sigmoid | 2.70 |
| w/o Link Regularization Loss | 2.64 |
| Perceptual Loss $\rightarrow$ MSE | 2.84 |
| PPL-1MemBank | 2.72 |

**Removal of hierarchy in the memory.** We introduce an ablated variant of PPL using only one memory bank, referred to as PPL-1MemBank. Below, we introduce its implementation. If each pose's keypoints were encoded into 34 vectors sharing the same embedding space, mean pooling these embeddings would not effectively distill the keypoint prior. Instead, PPL-1MemBank encodes all keypoints of a pose into a single vector, ensuring that all vectors share a common embedding space. To maintain comparable capacity, PPL-1MemBank uses a single memory bank with 512 vectors, equivalent to PPL's 34 memory banks with 16 vectors each. The keypoint prior is then obtained by mean pooling across this single memory bank. We evaluated PPL-1MemBank against PPL, with results summarized in the last row of **Table A3**. On the Human3.6m dataset, PPL-1MemBank

achieves a normalized L2 error of 2.72, compared to 2.56 with PPL's multi-bank approach. This indicates that multiple smaller memory banks improve pose estimation performance.

**PPL trained with randomly masked frames as reference images on video datasets.** We also train PPL on Human3.6m and Taichi by treating video frames as independent images and using randomly masked images as $I_{ref}$. We denote this setting as PPL*. The results are reported in **Table A4**. Notably, even with masked images as $I_{ref}$, PPL* consistently outperforms AutoLink. This highlights that the effectiveness of PPL stems primarily from its pose prior, rather than from cross-frame reconstruction.

Table A4: **Results of Training PPL* using the randomly selected and masked video frames as** $I_{ref}$ **on Human3.6M and Taichi video datasets.** We use resolution 256×256 for Taichi and 128×128 for Human3.6m. We use the evaluation metrics (↓) as reported in **Table 1**. Best is in bold.

| Methods | Human3.6m | Taichi |
|---------|-----------|--------|
| AutoLink | 2.76 | 316.10 |
| PPL (ours) | **1.92** | **293.35** |
| PPL* (ours) | 2.23 | 298.60 |

## A9    VISUALIZATION RESULTS ON SEMANTIC CONSISTENCY OF PREDICTED KEYPOINTS

Our model does not explicitly enforce semantic consistency across keypoints, as the learning process is fully unsupervised. However, we do observe partial consistency across instances—some keypoints tend to appear in similar regions, though they are not guaranteed to correspond to the same semantic part across different individuals. In **Figure A8**, for body parts with strong semantic distinctiveness—such as the head, neck, and ankles—our model consistently predicts the same keypoints across different examples.

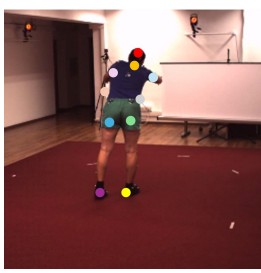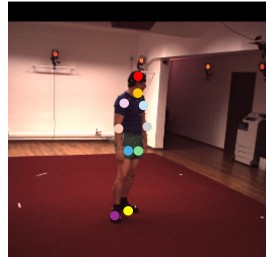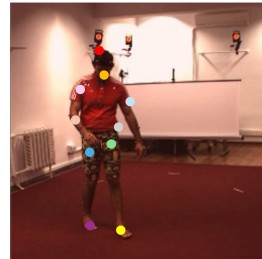

Figure A8: **Visualization results of our PPL on semantic consistency of the predicted keypoints on Human3.6m.** In each image, the color of each dot indicates its predicted keypoint index.

## A10    PRIOR LEARNING EXTENDS BEYOND POSE ESTIMATION: RECOGNITION EXPERIMENTS

To further clarify the distinction between prior learning and pose estimation, we conduct new experiments demonstrating that pose estimation serves only as a testbed, while the learned priors can naturally extend to other downstream tasks. Specifically, we extended our framework to visual recognition tasks that require different discriminative cues beyond human pose estimation. Specifically, we evaluated the prior-learning mechanism on Yoga82-Level1 Verma et al. (2020) (6 classes) and CIFAR-10 Krizhevsky et al. (2009) (10 classes) using ResNet-50 He et al. (2016) as the classification backbone.

In these experiments, the ResNet-50 model is trained only on clean images. For the prior-learning module, we train one PPL per class (6 PPLs for Yoga82-Level1 and 10 PPLs for CIFAR-10). At test

time, each occluded image is forwarded through all PPLs, where each PPL reconstructs the image according to its learned class-level prior and the prototypical poses in its memory. Each reconstructed output is then individually evaluated by the trained ResNet-50, which produces a classification logit. Among these logits, we select the one with the highest confidence as the final prediction. This inference process enables the system to recover discriminative visual features under occlusions using learned priors, without modifying the backbone classifier. Occlusions are introduced by randomly masking 50% of the image patches. The image is partitioned into $16 \times 16$ non-overlapping patches. The results are shown in the **Table A5**.

These results show that the learned priors provide benefits beyond pose estimation and transfer to recognition tasks with different forms of intra-class variability. The same PPL module can be integrated without altering the training objective or classification backbone, and consistently recovers accuracy under occlusions. This empirical evidence strengthens the claim that explicit prior learning serves as a general mechanism and is not limited to the specific task of pose estimation.

In the future, we plan to integrate PPL into more downstream tasks, such as context reasoning Cai et al. (2025); Liu et al. (2022); Ding et al. (2022), object discovery Wang et al. (2023), action recognition Han et al. (2024), scene graph generation Khandelwal et al. (2023), robotic navigation Piriyajitakonkij et al. (2024), continual learning Shi et al. (2025); Tee & Zhang (2023); Singh et al. (2023); Wu et al. (2023a); Talbot et al. (2023), and eye movement prediction Shi et al.; Jia et al. (2025); Wang et al. (2025); Zhang et al. (2017; 2022a).

Table A5: Classification accuracy (%) of ResNet-50 under clean and occluded settings, with and without PPL.

| Dataset | ResNet-50 (Clean) | ResNet-50 (Occluded) | ResNet-50 + PPL (Occluded) |
|---|---|---|---|
| Yoga82 Verma et al. (2020) | 94.6% | 76.5% | 88.2% |
| CIFAR-10 Krizhevsky et al. (2009) | 96.3% | 72.8% | 77.4% |

