# OpenReview forum: "Pose Prior Learner: Unsupervised Categorical Prior Learning for Pose Estimation"
_ICLR.cc/2026/Conference — ICLR 2026 Poster_

### Official Review · Reviewer_ap4i · 2025-10-20

**Soundness:** 4
**Presentation:** 3
**Contribution:** 3
**Rating:** 8
**Confidence:** 4

**Summary:**

This paper introduces the task of unsupervised categorical prior learning in pose estimation. The authors propose Pose Prior Learner, which is tasked with learning a general pose for a given category. The key design insight invloves the hierarchical memory that stores prototypical poses, from which the general pose is distilled. PPL does not require any annotated supervision, and its performance is improved after a few iterations of inferences.

**Strengths:**

1. The paper is written clearly.

2. Outperforms the baselines.

3. Design choices are clearly justified and explained. For example:

a. The hierarchical memory component is innovative and clearly justified in Section 3.3, through the 3 points mentioned.

b. In line 286: while the computation makes sense as a standard/reasonable regularizer, this interpretation gives it another justification.

4. The experiments conducted seem to convey about the effectiveness of the approach. Table 1 shows that the approach outperforms all other baselines. I appreciated lines 360-361 that cancel the temporal effect. In addition, the ablations in Table 2 are a good experiment showing the importance of finetuning the pose (especially with regards to human-defined priors).

**Weaknesses:**

1. The distinction between pose estimation and prior learning is nor clear to me. If you learn a pose, you implicitly learn a "pose prior". If there is an actual difference please clarify. If not, please consider stating that your approach estimate poses. This distinction challenged my understanding of what your approach does - it might be inferred that your learned prior will be used for pose estimation (lines 67-68 and lines 146-147).
Or, for example, in lines 70-76, why your approach (or an approach that predicts pose priors) cancels the risk of failure that you mention?

2. Teaser figure: the arrows over-complicate your presentation, and are more relevant to the actual solution and are less relevant to the challenge itself. Maybe simplifying the teaser figure by focusing on the task introduction would help. I don’t think the distinction between blue and red arrows are helpful, especially when it is not clear what is transformed or refined in the stage of observing this figure.

3. Hierarchical memory distillation is mentioned rather early and it is left underexpained untill section 3.3. First, it is suggested to hint earlier that further details are given in this section. Also, I would consider moving it as the beggining of Section 3, as this is the first component and seems like the key design contribution of the work.

4. Using a 2D CNN is considered out-dated in many use-cases. Did you consider using a transfomer-based frozen (or not frozen) foundation model as an image encoder?

5. In line 278 you mention that I_recon should be identical to I, and your loss do not really measure identity (if you did really want identity, why not use MSE which is lighter compared to perceptual loss). Also, can the boundary loss be also considered as regularization?

**Questions:**

Is there a semantic consistency between the predicted keypoints? That is, if keypoint i is in the head of the person, are we expected to see i on the heads of all other persons?

If so, maybe you should mention it, and visualize the keypoint numbers on top of at least one figure, so readers could see that the model learns semantic keypoints.

---

> ### Author Response · Authors · 2025-11-21
> **Response part 1**
>
> **Weakness.1:
> The distinction between pose estimation and prior learning is nor clear to me. If you learn a pose, you implicitly learn a "pose prior". If there is an actual difference please clarify. If not, please consider stating that your approach estimate poses. This distinction challenged my understanding of what your approach does - it might be inferred that your learned prior will be used for pose estimation (lines 67-68 and lines 146-147).
> Or, for example, in lines 70-76, why your approach (or an approach that predicts pose priors) cancels the risk of failure that you mention?**
>
> **Answer:**
> We would like to clarify how pose estimation and prior learning are fundamentally different in our formulation:
>
> __Prior learning is the core problem, pose estimation is only a testbed.__
> Our goal is to explore whether a categorical prior can be learned directly from raw visual data. Pose estimation is not the final goal, but rather a testbed used to examine whether the learned prior is meaningful and useful.
>
> __Learning poses does not imply learning a “pose prior.”__
> Existing methods (e.g., AutoLink) learn only an instance-wise distribution of keypoints. When a region is occluded, the model fails to detect keypoints in that region, which indicates that no implicit pose prior is actually learned to enforce completeness. Therefore, pose estimation alone does not guarantee the existence of a categorical prior.
>
> __Our method explicitly distills individual poses into a categorical prior.__
> During training, individual pose predictions and the prior evolve together. Initially, the prior is meaningless but individual poses can still converge. As these poses improve, the prior becomes more structured, and the increasingly meaningful prior in turn promotes more accurate pose estimation. This forms a mutual refinement process that enables knowledge to emerge beyond instance-level fitting.
>
> __The explicit prior ensures completeness and plausibility under occlusion.__
>  When input images are occluded, our learned prior restores the missing parts and enforces structural completeness. Additionally, the stored prototypical poses constrain the prediction to remain regular and appear in valid spatial configurations.
>
> For these reasons, we frame our approach as prior learning for pose estimation, not merely pose estimation with a stronger model. We revised the manuscript to clarify this distinction in __lines 055-057__ and __lines 097-104__ in the revised manuscript. We redesigned __Figure 1__ to more clearly illustrate the distinction and relationship between prior learning and pose estimation.
>
>
> **Weakness.2:
> Teaser figure: the arrows over-complicate your presentation, and are more relevant to the actual solution and are less relevant to the challenge itself. Maybe simplifying the teaser figure by focusing on the task introduction would help. I don’t think the distinction between blue and red arrows are helpful, especially when it is not clear what is transformed or refined in the stage of observing this figure.**
>
> **Answer:**
> Thank you for the suggestion. We agree that the current teaser figure may appear overly detailed. In the revised version, we simplified the figure to focus more on introducing the task, rather than illustrating the full solution. Specifically, __Figure 1__ is divided into two panels: The first presents the learning to capture the category-level pose prior from raw images. The second demonstrates the application of the learned prior during pose estimation on clean images and pose inference on occluded images.
>
> **Weakness.3:
> Hierarchical memory distillation is mentioned rather early and it is left underexpained untill section 3.3. First, it is suggested to hint earlier that further details are given in this section. Also, I would consider moving it as the beginning of Section 3, as this is the first component and seems like the key design contribution of the work.**
>
> **Answer:**
> Thank you for the helpful suggestion. We agree that hierarchical memory distillation is a key component of our method and should be introduced more clearly. In the revised version, we moved the hierarchical memory description to the beginning of Section 3, so the main contribution is presented upfront in a clearer manner (__line 208__). This restructuring should improve readability and clarify the role of this component in our framework.

---

> > ### Comment · Reviewer_ap4i · 2025-11-26
> >
> > I thank the authors for addressing most of my concerns. However, my primary concern remains unresolved.
> >
> > **"Prior learning is the core problem, pose estimation is only a testbed."** The core issue is that prior learning, not pose estimation, is presented as the central problem. However, when both training and evaluation are conducted exclusively on pose estimation tasks, the work effectively addresses pose estimation alone. While claiming a more general goal (pose prior learning) is a valid aspiration, it requires experimental validation beyond a single application domain.
> >
> > **Occlusion handling and implicit priors**: The authors correctly note that when a region is occluded, the model fails to detect keypoints in that region, suggesting that no implicit pose prior enforces completeness. They argue that pose estimation alone therefore does not guarantee learning a categorical prior. However, this argument has limitations: one could contend that a more effective pose estimation approach (supervised or unsupervised) would handle occlusions better. To substantiate the claim that "learning poses does not imply learning a pose prior," the authors need to evaluate whether a pose estimation model that handles occlusions effectively would necessarily learn a pose prior. In other words: would any high-performing pose estimation model (particularly regarding occlusion) inherently capture pose priors? Additionally, as the authors suggest, an effective pose prior approach handles pose estimation effectively. Therefore, when occlusions are handled correctly, are pose estimation and pose prior learning functionally equivalent?
> >
> > **Scope and motivation**: The distinction between unsupervised pose estimation and pose prior learning remains insufficiently justified, which challenges the understanding of the work's scope and objectives. The work appears to address pose estimation through a novel approach that explicitly models pose priors. I recommend the authors revise the motivation to reflect this framing more accurately.
> >
> > Despite these concerns, I maintain my positive evaluation of the work and retain my original rating.

---

> > > ### Author Response · Authors · 2025-11-29
> > >
> > > We thank the reviewer for the detailed and constructive comments. Below we respond to the remaining concerns.
> > >
> > > __1. Scope: Prior Learning Beyond Pose Estimation__
> > >
> > > To directly address this concern, we extended our framework to __visual recognition__ tasks that require different discriminative cues beyond human pose estimation. Specifically, we evaluated the prior-learning mechanism on __Yoga82-Level1 (6 classes)__ and __CIFAR-10 (10 classes)__ using ResNet-50 as the classification backbone.
> > >
> > > In these experiments, the ResNet-50 model is trained only on clean images. For the prior-learning module, we train one PPL per class (6 PPLs for Yoga82-Level1 and 10 PPLs for CIFAR-10). At test time, each occluded image is forwarded through all PPLs, where each PPL reconstructs the image according to its learned class-level prior and the prototypical poses in its memory. Each reconstructed output is then individually evaluated by the trained ResNet-50, which produces a classification logit. Among these logits, we select the one with the highest confidence as the final prediction. This inference process enables the system to recover discriminative visual features under occlusions using learned priors, without modifying the backbone classifier.
> > >
> > > Occlusions are introduced by randomly masking 50% of the image patches. The image is partitioned into 16 × 16 non-overlapping patches.
> > >
> > > The results are shown in the following table:
> > >
> > > | Dataset  | ResNet-50 (Clean) | ResNet-50 (Occluded) | ResNet-50 + PPL (Occluded) |
> > > |----------|-------------------|----------------------|-----------------------------|
> > > | Yoga82   | 94.6%             | 76.5%                | 88.2%                       |
> > > | CIFAR-10 | 96.3%             | 72.8%                | 77.4%                       |
> > >
> > > These results show that the learned priors provide benefits beyond pose estimation and transfer to recognition tasks with different forms of intra-class variability. The same PPL module can be integrated without altering the training objective or classification backbone, and consistently recovers accuracy under occlusions. This empirical evidence strengthens the claim that explicit prior learning serves as a general mechanism and is not limited to the specific task of pose estimation.
> > >
> > > __2. Relation Between Pose Estimation and Pose Prior Learning__
> > >
> > > We emphasize that handling occlusions does not necessarily imply that a pose prior has been learned. We list the cases where a model may perform well under occlusion by relying on local or temporal cues rather than storing categorical priors:
> > >
> > > 1. Local interpolation: Spatial smoothing or keypoint heatmap interpolation can recover occluded regions, but this only enforces local continuity—without any knowledge of global pose structure.
> > >
> > > 2. Temporal propagation: In video-based settings, a model may infer occluded keypoints by tracking them across neighboring frames. This succeeds under occlusion but merely relies on temporal continuity rather than a stored class-level prior.
> > >
> > > These cases show that a pose estimator can be robust under occlusion while still not learning a true pose prior. Therefore, pose estimation and prior learning remain related but not equivalent tasks.
> > >
> > > __3. Scope and Motivation__
> > >
> > > We appreciate the reviewer’s suggestion to refine the motivation. In the revised manuscript, we clarify that:
> > >
> > > 1. Our objective is not to replace pose estimation with prior learning, but to bridge them by showing how prior learning can handle occlusion cases that keypoint-based models struggle with.
> > >
> > > 2. The extended experiments on Yoga82-Level1 and CIFAR-10 support the same mechanism in broader visual recognition tasks, helping align the motivation with the empirical evidence.

---

> ### Author Response · Authors · 2025-11-22
> **Response part 2**
>
> **Weakness.4:
> Using a 2D CNN is considered out-dated in many use-cases. Did you consider using a transfomer-based frozen (or not frozen) foundation model as an image encoder?**
>
> **Answer:**
> We agree that transformer-based encoders are widely used. In our current work, we use a 2D CNN primarily because it allows controlled comparison with AutoLink, which also adopts a CNN-based encoder. Moreover, our focus is not on improving encoder performance, but on validating the feasibility of categorical prior learning. That said, our framework is not tied to CNN encoders. We mentioned this in the revised manuscript as a promising direction for future work (__line 539__).
>
> **Weakness.5:
> In line 278 you mention that I_recon should be identical to I, and your loss do not really measure identity (if you did really want identity, why not use MSE which is lighter compared to perceptual loss). Also, can the boundary loss be also considered as regularization?**
>
> **Answer:**
> When stating that I_recon, I_recon​  “should be identical” to I, we refer to semantic consistency, not strict pixel-wise identity. Therefore, we use a perceptual loss, which better captures structural and appearance-level information than MSE.
>
> To validate this choice, we conducted an additional experiment on Human3.6M by replacing the perceptual loss with MSE. The normalized L2 error increased from 2.56 to __2.84__, confirming that perceptual loss yields more accurate pose estimation.
>
> Regarding the boundary loss: yes, it can be considered a regularization term, as it encourages smoother predictions and helps the network converge faster.
>
> We integrated this clarification and the new result in the revised manuscript (__line 291__ and __lines 296-299__).
>
> **Question.1:**
> Is there a semantic consistency between the predicted keypoints? That is, if keypoint i is in the head of the person, are we expected to see i on the heads of all other persons?
> If so, maybe you should mention it, and visualize the keypoint numbers on top of at least one figure, so readers could see that the model learns semantic keypoints.
>
> **Answer:**
> Our model does not explicitly enforce semantic consistency across keypoints, as the learning process is fully unsupervised. However, we do observe partial consistency across instances—some keypoints tend to appear in similar regions, though they are not guaranteed to correspond to the same semantic part across different individuals.
>
> To address this suggestion, we added a visualization in __Appendix A9__ where keypoints predictions are overlaid on example images, allowing readers to inspect the semantic consistency of keypoints visually. We also briefly mentioned this consistency behavior in the main text (__lines 419-421__) to clarify our model’s capacity regarding semantic alignment.

---

### Official Review · Reviewer_9iwV · 2025-10-25

**Soundness:** 3
**Presentation:** 2
**Contribution:** 3
**Rating:** 6
**Confidence:** 4

**Summary:**

The paper suggests a new unsupervised pose estimation approach using learned pose priors.
It predicts a shared pose prior using a hierarchical memory module.
During inference, it iteratively predicts an affine transformation that adapts the priors to the input image, overcoming difficult scenarios like occlusions.
Training is done through image reconstruction and optimizing the memory bank.

**Strengths:**

- The paper suggests a new unsupervised pose estimation approach that can learn pose priors to overcome occlusions.
- The authors are willing to publish their code, model, and data.
- The method achieves SOTA performance on multiple datasets compared to similar-sized methods.

**Weaknesses:**

- The authors should better explain what is the difference between current unsupervised pose estimation methods and "categorical prior learning for pose estimation".
Unsupervised pose estimation learns pose estimation priors over the trained data distribution. It's not clear how this is different from the suggested framing (as described in contribution 1). If the suggested explicit way of describing the learned priors has an advantage over implicit ones, it should be justified.

**Questions:**

- I find the introduction a bit confusing - framing the suggested method as a prior learning problem, while the rest of the paper discusses pose estimation. I would like to better understand what the authors meant and how it differs from the latter.
- There is a discrepancy between line 917 and Table A3 regarding the performance using 1 memory bank.

---

> ### Author Response · Authors · 2025-11-21
>
> **Weakness.1:
> The authors should better explain what is the difference between current unsupervised pose estimation methods and "categorical prior learning for pose estimation". Unsupervised pose estimation learns pose estimation priors over the trained data distribution. It's not clear how this is different from the suggested framing (as described in contribution 1). If the suggested explicit way of describing the learned priors has an advantage over implicit ones, it should be justified.**
>
> **Answer:**
> While existing unsupervised pose estimation methods may implicitly encode structural priors, these priors remain hidden in network parameters and are not directly interpretable. Our method differs in two main aspects:
>
> __Explicit vs. implicit prior learning:__
>
> Prior methods learn pose priors implicitly as latent model weights, without exposing their structure. In contrast, our method explicitly extracts the pose prior and represents it through keypoint and connectivity priors. This explicit representation is symbolic, meaning it has a determinate and structured form that can be interpreted, visualized, and analyzed, rather than being an opaque byproduct of training.
>
> __Benefit of being symbolic__
>
> Because the prior has structured and interpretable form, it provides structural completeness and plausibility during inference, especially under challenging conditions like occlusion. This allows the model to infer missing body parts based on category-level structure, which implicit parameter-based priors cannot reliably achieve. For example, as shown in Figure A1 of Appendix A5, AutoLink lacks such a prior and consequently fails to predict the full body under occlusion, whereas our PPL successfully handles this case.
>
> In summary, our contribution lies in making the prior explicit and symbolic, enabling principled structural reasoning beyond what traditional implicit prior learning can offer. We provided this justification in __lines 097-104__ in the revised version.
>
>
> **Question.1:
> I find the introduction a bit confusing - framing the suggested method as a prior learning problem, while the rest of the paper discusses pose estimation. I would like to better understand what the authors meant and how it differs from the latter.**
>
> **Answer:**
> In our work, prior learning is the central objective, which we view as a new and important problem: can a categorical prior be learned directly from raw visual data? Pose estimation is not the goal, but rather a testbed to demonstrate the feasibility and usefulness of the learned prior. We redesigned __Figure 1__ to more clearly illustrate the distinction and relationship between prior learning and pose estimation.
>
> More importantly, this work proposes a new perspective on prior learning:
> we first estimate individual instances, and then summarize their common structure into a categorical prior that captures the shared configuration of the category. This shows that prior knowledge can naturally emerge from instance observations. We believe this perspective opens up a new direction for learning priors beyond pose estimation.
>
> **Question.2:
> There is a discrepancy between line 917 and Table A3 regarding the performance using 1 memory bank.**
>
> **Answer:**
> Thank you for pointing this out. This discrepancy is due to a typographical error. The correct value in the last row of Table A3 should be __2.72__. We corrected it in __line 962__ in the revised version.

---

> > ### Comment · Reviewer_9iwV · 2025-11-23
> >
> > I thank the authors for their detailed comment.
> > While some of my initial concerns were addressed, I still have a major concern remaining:
> >
> > The claim of general prior learning is not sufficiently proven or justified.
> > While the suggested method succeeds in suggesting an effective and explicit method of learning priors for pose estimation, I believe it should be framed as such.
> > Apart from the introduction, the whole paper is about pose estimation.
> > I think the authors should align the introduction with the rest of the paper, or alternatively, demonstrate the applicability of the suggested method to a domain other than pose estimation using another task.
> >
> > The paper would benefit from framing itself as "unsupervised pose estimation" rather than overreaching for goals that are not justified enough.

---

> > > ### Author Response · Authors · 2025-11-29
> > >
> > > We thank the reviewer for this valuable comment. To directly address this concern, we extended our framework to __visual recognition__ tasks that require different discriminative cues beyond human pose estimation. Specifically, we evaluated the prior-learning mechanism on __Yoga82-Level1 (6 classes)__ and __CIFAR-10 (10 classes)__ using ResNet-50 as the classification backbone.
> > >
> > > In these experiments, the ResNet-50 model is trained only on clean images. For the prior-learning module, we train one PPL per class (6 PPLs for Yoga82-Level1 and 10 PPLs for CIFAR-10). At test time, each occluded image is forwarded through all PPLs, where each PPL reconstructs the image according to its learned class-level prior and the prototypical poses in its memory. Each reconstructed output is then individually evaluated by the trained ResNet-50, which produces a classification logit. Among these logits, we select the one with the highest confidence as the final prediction. This inference process enables the system to recover discriminative visual features under occlusions using learned priors, without modifying the backbone classifier.
> > >
> > > Occlusions are introduced by randomly masking 50% of the image patches. The image is partitioned into 16 × 16 non-overlapping patches.
> > >
> > > The results are shown in the following table:
> > >
> > > | Dataset  | ResNet-50 (Clean) | ResNet-50 (Occluded) | ResNet-50 + PPL (Occluded) |
> > > |----------|-------------------|----------------------|-----------------------------|
> > > | Yoga82   | 94.6%             | 76.5%                | 88.2%                       |
> > > | CIFAR-10 | 96.3%             | 72.8%                | 77.4%                       |
> > >
> > > These results show that the learned priors provide benefits beyond pose estimation and transfer to recognition tasks with different forms of intra-class variability. The same PPL module can be integrated without altering the training objective or classification backbone, and consistently recovers accuracy under occlusions. This empirical evidence strengthens the claim that explicit prior learning serves as a general mechanism and is not limited to the specific task of pose estimation.

---

### Official Review · Reviewer_Q36a · 2025-10-26

**Soundness:** 3
**Presentation:** 3
**Contribution:** 3
**Rating:** 6
**Confidence:** 2

**Summary:**

The paper proposes to learn categorical keypoint pose prior via image reconstruction. It proposes a hierarchical memory mechanism to first learn the key point distribution into several different features space, and compute the keypoint prior by decoding the average features in the learned feature space. Then, the connectivity prior is learned jointly with the image reconstruction decoder while the keypoint prior is fixed. The image reconstruction results are refined iteratively during inference. Experiments are done on multiple datasets to demonstrate the effectiveness of this method.

**Strengths:**

1. Experiments are done on multiple datasets and the performance is strong, which proves the effectiveness of this method.

2. Ablation studies are done properly, showing how the keypoint prior and conectivity prior influences the performance.

3. Learning categorical prior is an interesting and important topic.

**Weaknesses:**

1. What is the downstream application of the learned prior? All experiments focus on image reconstruction, and since the prior is learned under the supervision of this task, it seems primarily tailored to it. However, it would be interesting to know whether the learned prior can generalize to or benefit other tasks beyond image reconstruction.

2. The evaluation protocol is not clearly described. Is the occluded image used as the input while the clean image serves as the ground truth? The paper should explicitly clarify this setting, along with the specific evaluation metrics used.

3. In Figure A5, the keypoints from AutoLink appear to have clearer semantic meaning compared to those extracted by the proposed method, especially around the upper body. Do the authors have insights into the semantic interpretation or consistency of the learned keypoint priors?

4. The evaluation of the keypoint prior learning process seems insufficient. Including qualitative visualizations of all learned keypoint templates, as well as the average learned prior, would provide valuable insight into the representational quality of the method.

**Questions:**

Please refer to the weaknesses. Overall, the method appears technically sound. My main concern lies in the lack of discussion or demonstration of potential downstream applications beyond image reconstruction. However, as I am not an expert in this area, I would defer to other reviewers’ opinions regarding the broader applicability and impact of the proposed approach.

---

> ### Author Response · Authors · 2025-11-21
>
> **Weakness.1:**
> What is the downstream application of the learned prior? All experiments focus on image reconstruction, and since the prior is learned under the supervision of this task, it seems primarily tailored to it. However, it would be interesting to know whether the learned prior can generalize to or benefit other tasks beyond image reconstruction.
>
> **Answer:**
> Thank you for the suggestion. We would like to clarify that image reconstruction is only used as the learning signal, not the ultimate goal of the learned pose prior. The prior is not tailored to reconstruction; instead, its purpose is to represent categorical structural information.
> Currently, we use unsupervised pose estimation as a testbed to examine how the learned prior can emerge and be utilized. We have already demonstrated its benefit for pose reasoning under occlusion, which goes beyond reconstruction. We redesigned __Figure 1__ to more clearly illustrate the distinction and relationship between prior learning and pose estimation.
>
> **Weakness.2:**
> The evaluation protocol is not clearly described. Is the occluded image used as the input while the clean image serves as the ground truth? The paper should explicitly clarify this setting, along with the specific evaluation metrics used.
>
> **Answer:**
> We use the summed or normalized L2 distance between the predicted and ground-truth keypoints as the evaluation metric, and we apply it consistently to both occluded and clean input images. For iterative inference on occluded images, the occluded image is directly used as input to estimate the poses, and the estimated pose is compared with the ground-truth keypoints. No clean image is used at this setting. We made this evaluation protocol explicit in the revised manuscript to avoid ambiguity in __lines 368-371__.
>
> **Weakness.3:**
> In Figure A5, the keypoints from AutoLink appear to have clearer semantic meaning compared to those extracted by the proposed method, especially around the upper body. Do the authors have insights into the semantic interpretation or consistency of the learned keypoint priors?
>
> **Answer:**
> We could not locate Figure A5, so we assume you are referring to Figure A1 in Appendix A5. We would like to clarify that both AutoLink and our method learn keypoints in a fully unsupervised manner, and neither method assigns semantic labels to keypoints during training.  However, we did observe that several keypoints in our learned pose prior consistently align with specific body parts. For example, in Figure A1, our method consistently predicts legs with reasonable length, whereas AutoLink fails to do so. This emergent consistency suggests that our model captures meaningful structural cues, indicating the correctness and reliability of our method. In the revised manuscript, we additionally provide visualizations of the estimated poses with their keypoints highlighted across sample frames (__Figure A8__). These visualizations show that the estimated keypoints consistently maintain the same semantic labels across different images (__lines 995-1000__).
>
> **Weakness.4:**
> The evaluation of the keypoint prior learning process seems insufficient. Including qualitative visualizations of all learned keypoint templates, as well as the average learned prior, would provide valuable insight into the representational quality of the method.
>
> **Answer:**
> We agree that visualizing the learned priors is valuable. However, visualizing every prototypical pose is not straightforward because our model does not use a single, flat global memory, but instead employs multiple hierarchical memory banks, each storing partial structures at different abstraction levels. Therefore, there is no one-to-one correspondence between a single “prototype” and a full pose template, making direct visualization of all prototypes impractical.

---

> ### Author Response · Authors · 2025-11-29
> **Downstream Application of The Learned Prior: New Recognition Experiments**
>
> To further clarify the distinction between prior learning and pose estimation, we conduct new experiments demonstrating that pose estimation serves only as a testbed, while the learned priors can naturally extend to other downstream tasks. We extended our framework to __visual recognition__ tasks that require different discriminative cues beyond human pose estimation. Specifically, we evaluated the prior-learning mechanism on __Yoga82-Level1 (6 classes)__ and __CIFAR-10 (10 classes)__ using ResNet-50 as the classification backbone.
>
> In these experiments, the ResNet-50 model is trained only on clean images. For the prior-learning module, we train one PPL per class (6 PPLs for Yoga82-Level1 and 10 PPLs for CIFAR-10). At test time, each occluded image is forwarded through all PPLs, where each PPL reconstructs the image according to its learned class-level prior and the prototypical poses in its memory. Each reconstructed output is then individually evaluated by the trained ResNet-50, which produces a classification logit. Among these logits, we select the one with the highest confidence as the final prediction. This inference process enables the system to recover discriminative visual features under occlusions using learned priors, without modifying the backbone classifier.
>
> Occlusions are introduced by randomly masking 50% of the image patches. The image is partitioned into 16 × 16 non-overlapping patches.
>
> The results are shown in the following table:
>
> | Dataset  | ResNet-50 (Clean) | ResNet-50 (Occluded) | ResNet-50 + PPL (Occluded) |
> |----------|-------------------|----------------------|-----------------------------|
> | Yoga82   | 94.6%             | 76.5%                | 88.2%                       |
> | CIFAR-10 | 96.3%             | 72.8%                | 77.4%                       |
>
> These results show that the learned priors provide benefits beyond pose estimation and transfer to recognition tasks with different forms of intra-class variability. The same PPL module can be integrated without altering the training objective or classification backbone, and consistently recovers accuracy under occlusions. This empirical evidence strengthens the claim that explicit prior learning serves as a general mechanism and is not limited to the specific task of pose estimation.

---

### Official Review · Reviewer_vF1h · 2025-11-01

**Soundness:** 2
**Presentation:** 2
**Contribution:** 2
**Rating:** 2
**Confidence:** 4

**Summary:**

The paper introduces the challenge of unsupervised categorical prior learning in pose estimation and proposes PPL to learn the general pose prior for any object category. The core novelty is to use a hierarchical memory to store the compositional parts of prototypical poses for distillation, achieving improved pose estimation accuracy through template transformation and image reconstruction. The experiments show their effectiveness for pose estimation from occluded images.

**Strengths:**

1. The paper introduces the challenge of unsupervised categorical prior learning for human/animal pose estimation.
2. The proposed method presents an iterative strategy to progressively refine the estimated poses to the nearest prototypical poses in memory, showing superior performance even in occluded scenes.

**Weaknesses:**

1. Overstated novelty: The paper presents unsupervised categorical prior learning for pose estimation as a new challenge, but this novelty is not new and has also been addressed in category-level object pose estimation methods with learnable pose priors in self-supervised manner, like in [1][2]. Also, the related work needs to include these categorical prior learning methods to avoid any misleading.
2. Insufficient quantitative evaluation on non-human data. Although the authors claim that their approach generalizes beyond human datasets, only qualitative visualizations are provided in the appendix. More quantitative evaluations on diverse non-human datasets are necessary to support this generalization claim and should be included in the main manuscript.
3. Lack of ablation studies on method components. The paper heavily builds on AutoLink [3] and introduces several modules and four loss terms for optimization, but no ablation experiments are conducted to evaluate their individual contributions. This is crucial to demonstrate the effectiveness and necessity of each proposed design component. I am also interested in both the training and inference efficiency of the proposed method compared to AutoLink. Will the prior learning affect the computational efficiency compared to the prior-free method?
4. The layout and the writing of the paper need to be improved. For example, Table 1 needs to be placed on page 7, Figure 2 and Figure 3 also need to be rearranged to appear near the related content.
5. Lack of description for the used metrics for pose estimation. It is also recommended to include directional arrows for each metric to indicate whether higher or lower values denote better performance.
6. The comparison set includes relatively old methods. To ensure a fair and up-to-date evaluation, more recent approaches such as [4][5] need to be included in the comparison.

[1] Chen, Xu, et al. "Category-level object pose estimation via neural analysis-by-synthesis." European Conference on Computer Vision. Cham: Springer International Publishing, 2020.

[2] Guo, Jiaxin, et al. "A visual navigation perspective for category-level object pose estimation." European Conference on Computer Vision. Cham: Springer Nature Switzerland, 2022.

[3] He, Xingzhe, Bastian Wandt, and Helge Rhodin. "Autolink: Self-supervised learning of human skeletons and object outlines by linking keypoints." Advances in Neural Information Processing Systems 35 (2022): 36123-36141.

[4] Hedlin, Eric, et al. "Unsupervised keypoints from pretrained diffusion models." Proceedings of the IEEE/CVF Conference on Computer Vision and Pattern Recognition. 2024.

[5] Wang, Dongkai, et al. "Generalizable Object Keypoint Localization from Generative Priors." Proceedings of the Computer Vision and Pattern Recognition Conference. 2025.

**Questions:**

Please refer to the weaknesses. Overall, I feel that the paper is not ready for publication, due to its overstated contribution, insufficient experiments, and limited writing clarity.

---

> ### Author Response · Authors · 2025-11-21
> **Response part 1**
>
> **Weakness.1:
> Overstated novelty: The paper presents unsupervised categorical prior learning for pose estimation as a new challenge, but this novelty is not new and has also been addressed in category-level object pose estimation methods with learnable pose priors in self-supervised manner, like in [1][2]. Also, the related work needs to include these categorical prior learning methods to avoid any misleading.**
>
> **Answer:**
> We included [1,2] in the related work. However, our formulation is fundamentally different from these category-level pose estimation methods in the following aspects:
>
> __Learning goal:__ Methods [1,2] assume that objects within one category share a canonical pose and aim to estimate individual poses for that category. In contrast, our framework does not focus on category-level pose estimation. Instead, we learn priors of categorical structure from raw data, and pose estimation emerges only as a byproduct. While categorical pose priors implicitly exist in existing pose-estimation models, they are never interpreted, visualized, or treated as an independent entity from the hierarchical memory that can be used independently, whereas our work makes the prior explicit and usable.
>
> __Learning condition:__ Methods [1,2] are not unsupervised — they commonly require ground-truth poses, CAD models, or reconstruction constraints. Our approach is fully unsupervised, relying solely on raw image sequences without any labels or ground-truth poses.
>
> __Output:__ Existing methods output estimated individual poses only, without an explicit prior.
> In contrast, our model additionally outputs a learned categorical prior, which can be directly visualized and used to infer poses even under occlusion — a capability not explored in previous works.
>
> We emphasized these differences in __lines 138 - 148__ in the revised manuscript.
>
> **Weakness.2:
> Insufficient quantitative evaluation on non-human data. Although the authors claim that their approach generalizes beyond human datasets, only qualitative visualizations are provided in the appendix. More quantitative evaluations on diverse non-human datasets are necessary to support this generalization claim and should be included in the main manuscript.**
>
> **Answer:**
> We agree that quantitative evaluation on non-human data is valuable.
> Indeed, we provided quantitative results on CUB-200-2011 in Table 1, showing strong performance on a non-human dataset.
> For the remaining non-human datasets introduced in the paper, ground truth pose annotations are not available, because these datasets were originally not designed for pose estimation tasks and therefore contain no ground-truth pose labels. As a result, quantitative evaluation is infeasible.
>
> We thus provided qualitative visualizations for these datasets, which consistently demonstrate the emergence of stable and meaningful poses across various animate and non-animate categories. We plan to curate such annotated datasets and provide their quantitative results as part of our future work. This discussion has now been added to the future work section in __lines 536-539__.

---

> ### Author Response · Authors · 2025-11-22
> **Response part 2**
>
> **Weakness.3:
> Lack of ablation studies on method components. The paper heavily builds on AutoLink [3] and introduces several modules and four loss terms for optimization, but no ablation experiments are conducted to evaluate their individual contributions. This is crucial to demonstrate the effectiveness and necessity of each proposed design component. I am also interested in both the training and inference efficiency of the proposed method compared to AutoLink. Will the prior learning affect the computational efficiency compared to the prior-free method?**
>
> **Answer:**
> We appreciate the reviewer’s suggestion and agree that ablation studies are important for evaluating component effectiveness. However, our paper already provides systematic ablations covering the key aspects of the proposed method:
>
> __Ablation on prior settings:__
> Table 2 presents detailed ablation results on different prior configurations, clearly demonstrating the necessity of each design choice related to categorical prior learning.
>
> __Ablation on loss components:__
> Table A3 includes ablation experiments on the four loss terms, showing their individual contributions to the final performance.
>
> __Regarding modules inherited from AutoLink:__
> Our primary contribution lies in developing a model for categorical prior learning, rather than re-evaluating components already validated in AutoLink. As such, ablating AutoLink’s original modules would not yield new insights and is beyond the scope of our work.
>
> Regarding computational efficiency, our hierarchical memory and prior extraction modules are lightweight with about __2.4M__ parameters only and does not introduce significant training overhead. This is compared against most of the large diffusion models which use pre-trained large language models as priors (usually over 900M parameters). Furthermore, computational efficiency is not the main objective of our method. Instead, our work primarily aims to study categorical prior learning and its effectiveness for pose reasoning under occlusion.
>
> **Weakness.4:
> The layout and the writing of the paper need to be improved. For example, Table 1 needs to be placed on page 7, Figure 2 and Figure 3 also need to be rearranged to appear near the related content.**
>
> **Answer:**
> Thank you for pointing this out. We rearranged __Table 1__, __Figure 2__, and __Figure 3__ to ensure they appear closer to the relevant content. We also improved the overall layout in the revised version.
>
> **Weakness.5:
> Lack of description for the used metrics for pose estimation. It is also recommended to include directional arrows for each metric to indicate whether higher or lower values denote better performance.**
>
> **Answer:**
> Note that we use only one metric L2 error, throughout all the experiments. For better clarity, we have now added directional arrows in the caption of __Table 1, 2, A2, A3__, and __A4__ and included a short summary of the metrics in the main text (__lines 368-370__) to make their interpretation more straightforward.
>
> **Weakness.6:
> The comparison set includes relatively old methods. To ensure a fair and up-to-date evaluation, more recent approaches such as [4][5] need to be included in the comparison.**
>
> **Answer:**
> We highlight the differences between the referenced works and our work as follows:
>
> __Regarding [4]:__
> A quantitative comparison with [4] has already been provided in Appendix A4. We note that [4] relies on a pre-trained conditional Stable Diffusion model with additional textual modality, making the comparison less fair under our purely visual and unimodal setting.
>
> __Regarding [5]:__
> This work tackles a different task and uses datasets that do not overlap with ours. Moreover, it focuses on text-guided generative modeling, whereas our work centers on unsupervised prior learning for pose estimation using visual inputs only, making direct comparison inappropriate.
>
> We clarified these points more explicitly in the revised manuscript (__lines 781-792__).
>
> [1] Chen, Xu, et al. "Category-level object pose estimation via neural analysis-by-synthesis." European Conference on Computer Vision. Cham: Springer International Publishing, 2020.
>
> [2] Guo, Jiaxin, et al. "A visual navigation perspective for category-level object pose estimation." European Conference on Computer Vision. Cham: Springer Nature Switzerland, 2022.
>
> [3] He, Xingzhe, Bastian Wandt, and Helge Rhodin. "Autolink: Self-supervised learning of human skeletons and object outlines by linking keypoints." Advances in Neural Information Processing Systems 35 (2022): 36123-36141.
>
> [4] Hedlin, Eric, et al. "Unsupervised keypoints from pretrained diffusion models." Proceedings of the IEEE/CVF Conference on Computer Vision and Pattern Recognition. 2024.
>
> [5] Wang, Dongkai, et al. "Generalizable Object Keypoint Localization from Generative Priors." Proceedings of the Computer Vision and Pattern Recognition Conference. 2025.

---

> ### Author Response · Authors · 2025-11-29
> **Prior Learning Extends Beyond Pose Estimation: New Recognition Experiments**
>
> To further clarify the distinction between prior learning and pose estimation, we conduct new experiments demonstrating that pose estimation serves only as a testbed, while the learned priors can naturally extend to other downstream tasks. We extended our framework to __visual recognition__ tasks that require different discriminative cues beyond human pose estimation. Specifically, we evaluated the prior-learning mechanism on __Yoga82-Level1 (6 classes)__ and __CIFAR-10 (10 classes)__ using ResNet-50 as the classification backbone.
>
> In these experiments, the ResNet-50 model is trained only on clean images. For the prior-learning module, we train one PPL per class (6 PPLs for Yoga82-Level1 and 10 PPLs for CIFAR-10). At test time, each occluded image is forwarded through all PPLs, where each PPL reconstructs the image according to its learned class-level prior and the prototypical poses in its memory. Each reconstructed output is then individually evaluated by the trained ResNet-50, which produces a classification logit. Among these logits, we select the one with the highest confidence as the final prediction. This inference process enables the system to recover discriminative visual features under occlusions using learned priors, without modifying the backbone classifier.
>
> Occlusions are introduced by randomly masking 50% of the image patches. The image is partitioned into 16 × 16 non-overlapping patches.
>
> The results are shown in the following table:
>
> | Dataset  | ResNet-50 (Clean) | ResNet-50 (Occluded) | ResNet-50 + PPL (Occluded) |
> |----------|-------------------|----------------------|-----------------------------|
> | Yoga82   | 94.6%             | 76.5%                | 88.2%                       |
> | CIFAR-10 | 96.3%             | 72.8%                | 77.4%                       |
>
> These results show that the learned priors provide benefits beyond pose estimation and transfer to recognition tasks with different forms of intra-class variability. The same PPL module can be integrated without altering the training objective or classification backbone, and consistently recovers accuracy under occlusions. This empirical evidence strengthens the claim that explicit prior learning serves as a general mechanism and is not limited to the specific task of pose estimation.

---

### Author Response · Authors · 2025-11-21
**General rebuttal**

We thank the reviewers for feedback and suggestions. We present the figures in PDF and the responses to individual reviewers’ questions below. The original questions from the reviewers are summarized and bolded.

---

### Meta-Review · Area_Chair_tw9D · 2026-01-05

**Summary:**

Most of the concerns raised by reviewers have been addressed by the rebuttal. However, while the paper claims that the proposed approach generalises beyond human datasets, quantitative evaluations on diverse non-human datasets are missing. In addition, almost all reviewers raised the same concern on the clarification about the distinction between prior learning and pose estimation. However the rebuttal still fails to well justified this. More importantly, most of the experiments are about pose estimation as a testbed while the paper claims that priors can be beneficial in many tasks but the applicability of the proposed method to other domains other than pose estimation is still not sufficiently addressed. However, considering positive feedback from three reviewers, and considering the fact that several key experiments were overlooked by the Reviewer vF1h (mainly the content of the supp file), the AC recommends acceptance.

**Reviewer Concerns:**

Most of the concerns have been addressed by the rebuttal and almost all reviewers (except vF1h) are positive about this work.

However, while the paper claims that the proposed approach generalises beyond human datasets, quantitative evaluations on diverse non-human datasets are missing. In addition, almost all reviewers raised the same concern on the clarification about the distinction between prior learning and pose estimation. However the rebuttal still fails to well justified this. More importantly, most of the experiments are about pose estimation as a testbed while the paper claims that priors can be beneficial in many tasks but the applicability of the proposed method to other domains other than pose estimation is still not sufficiently addressed.

**Reviewer Scores:**

I think Reviewer vF1h would increase its score to 4 as they overlooked some key experiments. Other reviewers probably keep their scores.

---

### Decision · Program_Chairs · 2026-01-26

Accept (Poster)